# Towards Enhanced Image Generation via Multi-Modal Chain of Thought in Unified Generative Models

## Abstract

Unified generative models have shown remarkable performance in both text and image generation. When faced with image synthesis tasks, they adopt straightforward text-to-image (T2I) generation. However, we find that direct T2I generation limits unified generative models in handling complex compositional instructions. Such instructions frequently occur in realistic application scenarios. Although this is a vital issue, existing works predominantly focus on improving the basic image generation capability of unified generative models. While improvements in basic image generation can contribute to complex image generation to some extent, they still fail to adequately resolve the problem. Inspired by Chain of Thought (CoT) solving complex problems in a step-by-step manner, this work aims to introduce CoT into unified generative models to address the challenges of complex image generation that direct T2I generation cannot effectively solve, thereby endowing models with enhanced image generation ability. To achieve this, we first introduce Functionality-oriented eXperts (FoXperts), an expert-parallel architecture in our model **FoX**, which assigns experts based on function. In this way, FoXperts disentangles the potential conflicts in current mainstream modality-oriented designs and provides a sound foundation for CoT. When introducing CoT, the first question is how to design a CoT approach specifically for complex image generation. To this end, we emulate a human-like artistic workflow—**planning, acting, reflection, and correction**—and propose the **Multimodal Chain of Thought (MCoT)** approach, since the data here involves multiple modalities (text and image). In response to the subsequent challenge—how to design an effective MCoT training paradigm—we develop a multi-task joint training paradigm that equips the model with all capabilities required for each MCoT step in a disentangled manner. This paradigm overcomes the difficulty and impracticality of collecting consistent multi-step data tuples for training. Extensive experiments demonstrate that **FoX** consistently outperforms existing unified models on various T2I benchmarks, delivering notable quantitative improvements in complex image generation.

## 1 Introduction

Unified generative models (Kondratyuk et al., 2023; He et al., 2024; Bachmann et al., 2024; Zhou et al., 2024b), particularly GPT-5 (OpenAI, 2025), have recently demonstrated superior capabilities in understanding and generating multimodal information, *e.g.*, linguistic and visual data. On image generation tasks, these models utilize straightforward text-to-image (T2I) generation and often achieve comparable or even superior performance to pure diffusion models (Rombach et al., 2022; Podell et al., 2024; Esser et al., 2024a). However, when faced with complex compositional instructions such as multi-object co-occurrence, attribute binding, or spatial constraints, straightforward T2I generation limits the performance of unified generative models. As illustrated in the first row of Figure 1, results produced by our model FoX with direct T2I generation reveal several limitations, including concept confusion in multi-object scenarios, failures even when only two objects are involved, attribute errors such as misaligned color binding, spatial inconsistencies like incorrect positioning, and object defects such as incomplete structures.

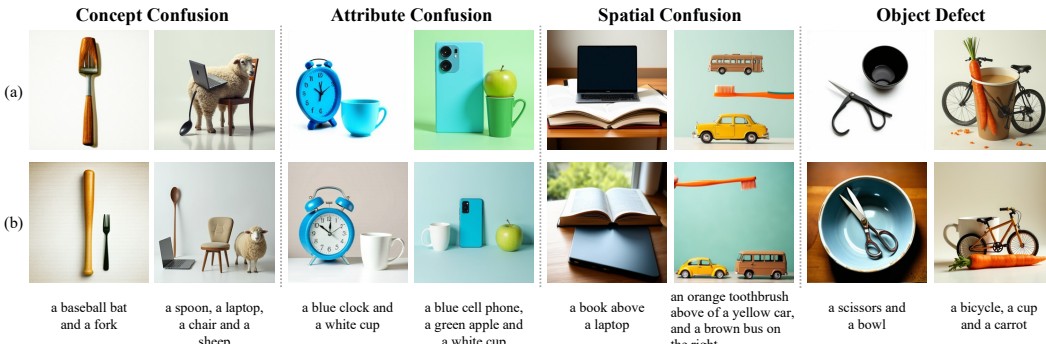

Figure 1: Limitations of straightforward text-to-image (T2I) generation. (a) The results generated by FoX with T2I generation illustrate confusion and defects. (b) The results generated by FoX with MCoT effectively address these issues.

Although such complex compositional instructions frequently occur in realistic application scenarios and constitute a vital issue, existing works predominantly focus on improving the basic image generation capability of unified generative models. In general, prior works mainly improve basic image generation ability by refining model architectures. Prior works (Ding et al., 2021; Team, 2024b; Lu et al., 2024; Kondratyuk et al., 2023) employ next-token prediction frameworks that represent all modalities, including visual data, as discrete tokens. However, such discrete tokenization approaches do not align with the continuous nature of images and videos, thereby limiting the visual generative potential. To address these limitations, subsequent studies (Zhou et al., 2024b; Xiao et al., 2024; Ma et al., 2024) explore hybrid generative architectures that treat text as discrete tokens in an autoregressive manner while processing images as continuous signals via diffusion. Furthermore, recent works design parallel experts assigned to modalities, namely modality-oriented architectures (Shi et al., 2024; Liang et al., 2024). While improvements on basic image generation, introduced by such architectural iterations, may contribute to complex image generation to some extent, they still fail to adequately resolve the problem.

Inspired by Chain of Thought (CoT) (Wei et al., 2022) effectively solving complex problems, we introduce CoT into unified generative models to similarly address complex image generation. Since CoT decomposes a complex task into simpler steps that can be more easily handled, it naturally suited for this challenge that direct T2I generation struggles with. To support CoT, we propose Functionality-oriented eXperts (FoXperts) as the architecture of our model FoX (Figure 2.a), designed to enhance fundamental visual understanding and generation capabilities, thereby providing a solid foundation for CoT in image generation. Unlike modality-oriented architectures that use a single shared visual expert for both tasks, FoXperts assign experts by function, dedicating separate experts to understanding and generation. This design disentangles potential functional conflicts in a single shared expert, which is otherwise jointly optimized for fundamentally different objectives—understanding (comprehension loss) and generation (diffusion loss).

Building on this, we introduce CoT, but another question arises: how to design a CoT approach specifically for complex image generation. Currently, CoT methods can be categorized into two types. One type is learned entirely through end-to-end training (Guo et al., 2025a; Lightman et al., 2023), which cannot be adopted here due to the lack of multi-step data for training CoT in image generation. The other type follows a human-defined key steps manner, where complex tasks are decomposed into human-defined subproblems (Zhou et al., 2023; Jiang et al., 2024; Mitra et al., 2024; Xu et al., 2024; Guo et al., 2025b; Zhao et al., 2025), but these are mainly designed for specific tasks and are not compatible with our scenario. Therefore, we follow the second manner and propose our **Multimodal Chain of Thought (MCoT)** approach for complex image generation, as the data involves multiple modalities (text and image). MCoT consists of four key steps—**planning, acting, reflection, and correction**—whose stepwise decomposition emulates a human artistic workflow, as shown in Figure 2.(b).

Furthermore, in response to the subsequent challenge of designing an effective MCoT training paradigm, we propose a multi-task joint training approach. This paradigm disentangles end-to-end training into separate subtasks, avoiding the need for consistent multi-step data tuples. As illustrated

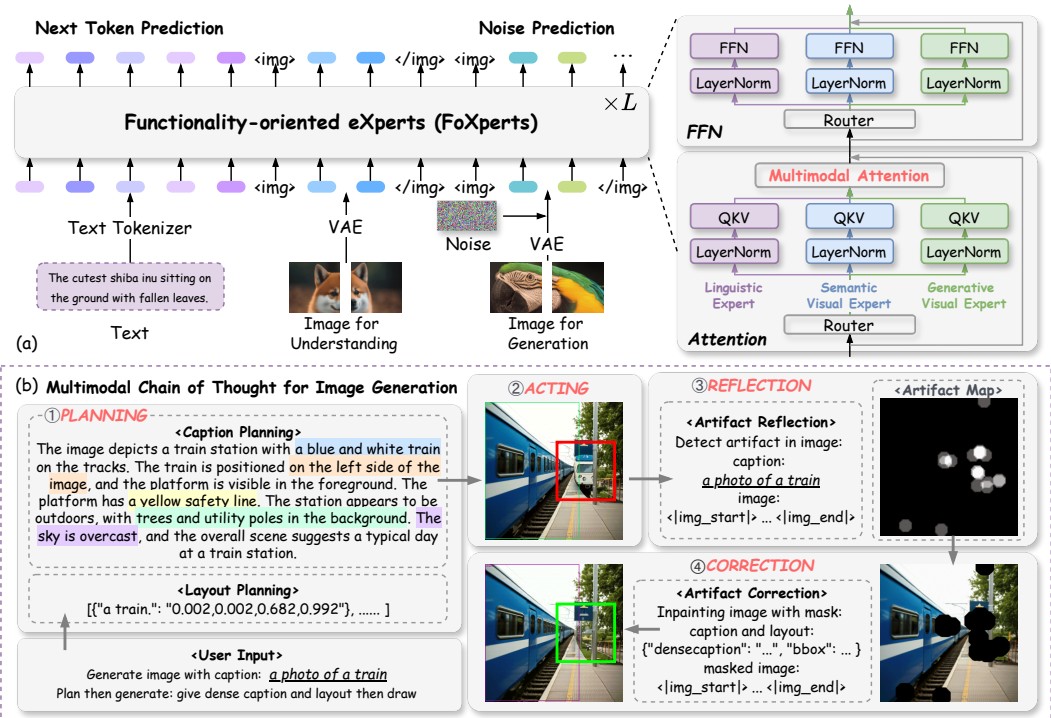

Figure 2: Overall Framework for FoX. (a) The illustration of architecture, highlighting FoXperts as the core component. (b) The illustration of MCoT approach for enhanced image generation.

in Figure 2.(b), a complete tuple includes {input prompt, detailed planning caption, layout planning boxes, first "wrong" image, artifact map, final "correct" image}. While available datasets provide only prompt-image pairs, we can use VLMs and detection models to generate planning captions and layout boxes. It is highly challenging to create the first "wrong" image that consistently aligns with the prompt, planning, and final image, while containing realistic errors for subsequent reflection and correction steps as shown in Figure 2.(b). Therefore, we avoid this obstacle by training the model in a disentangled manner, where data for each step is easily obtainable.

In summary, our main contributions are as follows:

- We introduce FoX, a unified generative model with a Functionality-oriented eXperts architecture that alleviates function-domain conflicts in modality-oriented designs, while seamlessly integrating both understanding and generation across textual and visual modalities.

- We propose a MCoT approach for enhanced image generation, together with a multi-task joint training paradigm that activates MCoT capabilities in a disentangled manner, thereby overcoming the challenge of collecting consistent multi-step data tuples.

- FoX exhibits explicit reasoning capability for image generation, enabling effective decomposition of complex problems and achieving improved performance across diverse benchmarks, including GenEval, T2I-CompBench, MS-COCO, VQA-v2, MME, and MMBench.

## 2 RELATED WORKS

**Diffusion Models.** Recent advancements in diffusion models have been remarkable, with notable contributions from the Stable Diffusion (SD) series (Rombach et al., 2022; Podell et al., 2024; Esser et al., 2024a), DALL-E (Ramesh et al., 2022; Betker et al., 2023b), and Imagen (Ho et al., 2022). These models are primarily developed for text-to-image generation. Subsequent efforts such as ControlNet (Zhang et al., 2023a), T2I-Adapter (Mou et al., 2024), and StyleShot (Gao et al., 2024) further extend their controllability and adaptability. However, diffusion models remain confined to image generation and lack broader multimodal generative capabilities.

**Generative Foundation Models.** In natural language generation, the GPT series (OpenAI, 2023) has shown that large language models (LLMs) can acquire broad task-solving abilities through large-scale training. Extending beyond language, vision language models (VLMs) (Liu et al., 2023; Chen et al., 2024; Bai et al., 2025) integrate vision and language understanding but lack image generation capability. To address this, some works couple LLMs with diffusion models (Sun et al., 2024a; Ge et al., 2024b; Wu et al., 2024b), while others employ discrete tokens to support both text and image generation (Team, 2024a; Lu et al., 2024), though with limited image quality. Hybrid generative models (Zhou et al., 2024a; Xie et al., 2024b) further unify autoregressive text generation and diffusion-based image generation. More recently, methods such as MoT (Liang et al., 2024) and LMFusion (Shi et al., 2024) propose modality-oriented architectures that assign experts to each modality, thus alleviating modality conflicts and improving text and image generation performance. However, such a design still suffers from function-domain conflicts. While methods like BAGEL (Deng et al., 2025) attempt to mitigate function-domain conflicts but instead introduce modality conflicts (by placing text and image modalities within the same branch), leading to performance degradation, as empirically evidenced in BAGEL's backbone work MoT. In response to this, we propose FoXperts, which maintains complete text–image modality separation while additionally introducing function-specific separation for image understanding and generation.

**Chain of Thought (CoT).** CoT (Wei et al., 2022) improves model performance on complex tasks by enabling step-by-step reasoning. It has been extensively studied in language models (Wei et al., 2022; Geva et al., 2021; Lightman et al., 2023; Guo et al., 2025a) and increasingly adopted in other domains (Mitra et al., 2024; Zhao et al., 2025), yet remains underexplored in image generation. Existing CoT approaches can be broadly categorized into two types: (1) Learned entirely end-to-end, where reasoning ability emerges purely from training (e.g., DeepSeek-R1 (Guo et al., 2025a), OpenAI o1 (Lightman et al., 2023)), which cannot be adopted here due to the lack of multi-step data for end-to-end training CoT in image generation; and (2) Human-defined for key steps, where complex tasks are decomposed into human-defined key subproblems, as in (Zhou et al., 2023; Jiang et al., 2024; Mitra et al., 2024; Xu et al., 2024; Guo et al., 2025b; Zhao et al., 2025), covering LLMs, VLMs, vision-language-action models (VLA), and large multimodal models (LMMs), which are mainly designed for specific tasks and are not compatible with our scenario. Previous human-defined CoT works such as CoT-VLA (Zhao et al., 2025) are specifically designed for robotic tasks, rendering them fundamentally incompatible with our complex image generation setting. Recently, several human-defined CoT methods for image generation are proposed, such as T2I-R1 (Jiang et al., 2025) and GoT-R1 (Duan et al., 2025). Although these approaches utilize the planning mechanism to enhance generation capabilities, they overlook a critical challenge in complex image generation: models equipped with the planning mechanism may still fail to render all image components perfectly in a single generation attempt, leaving residual image defects as illustrated in Figure 3. In response to this, we enhance the planning pipeline by integrating our reflection-correction mechanism, which alleviates residual image defects and thus leads to performance improvements on complex image generation benchmarks, as empirically validated in Sec. 4.5.

# 3 METHOD

## 3.1 PRELIMINARIES

**Language Modeling.** Let $z = (z_1, \ldots, z_N) \in V^N$ denote a sequence of discrete tokens drawn from a vocabulary $V$. Autoregressive language models factorize the joint distribution of these tokens using a causal decomposition:

$$P(z) = \prod_{i=1}^{N} P_\theta(z_i \mid z_{<i}), \tag{1}$$

where $\theta$ parameterizes the conditional distributions. This factorization allows the model to predict the next token based on all previously generated tokens. Training of the language model aims to minimize the negative log-likelihood over the dataset:

$$\mathcal{L}_{\text{LM}} = \mathbb{E}_{z \sim \mathcal{D}} \left[ -\sum_{i=1}^{N} \log P_\theta(z_i \mid z_{<i}) \right]. \tag{2}$$

For multimodal data, such as image-caption pairs $(x, c)$, the model conditions each token not only on previous tokens $z_{<i}$ but also on visual features extracted from the image: $P_\theta(z_i \mid z_{<i}, \phi(x))$, where $\phi(\cdot)$ denotes the visual encoder.

Rectified Flow defines a deterministic generative process for image data. Given images $\mathbf{x} \sim \mathcal{D}$ and Gaussian noise $\epsilon \sim \mathcal{N}(0, I)$, we construct linear trajectories between noise and data:

$$\mathbf{x}_t = t\mathbf{x} + (1 - t)\epsilon, \quad t \in [0, 1]. \tag{3}$$

The velocity field model $v_\theta$ predicts straightening directions through:

$$\mathcal{L}_{\text{RF}} = \mathbb{E}_{t \sim U(0,1), \mathbf{x}, \epsilon, c} \left[ \|(\mathbf{x} - \epsilon) - v_\theta(\mathbf{x}_t, t, c)\|_2^2 \right], \tag{4}$$

which contrasts with the stochastic differential equations used in Denoising Diffusion Probabilistic Models (DDPM) by employing deterministic optimal transport paths.

## 3.2 INPUT REPRESENTATION

Text inputs $T$ are transformed into an embedding sequence $x_{\text{text}} \in \mathbb{R}^{L \times d}$ using Qwen2's tokenizer (Bai et al., 2023), where $L$ is the sequence length and $d$ s the embedding dimension. An image $I \in \mathbb{R}^{H \times W \times 3}$, with height $H$ and width $W$, is encoded into a latent representation using SD3's Variational Autoencoder (VAE) (Esser et al., 2024b). Following the approach from Transfusion (Zhou et al., 2024b), we compress $2 \times 2$ patches into a single vector, resulting in image tokens $x_{\text{image}} \in \mathbb{R}^{\frac{H}{16} \times \frac{W}{16} \times d}$ after linear projection, where each token corresponds to a $16 \times 16$ pixel patch of the original image. We use the same VAE for image understanding tasks, consistent with prior works (TransFusion (Zhou et al., 2024b), LMFusion (Shi et al., 2024)). It preserves fine-grained, structurally rich visual features to facilitate detailed perception and may benefit MCoT steps like the reflection phase. To accommodate and differentiate the newly introduced image representations, we have expanded the original vocabulary of Qwen2, which consists of 151,936 entities, by incorporating 6 functional special tokens.

## 3.3 FUNCTIONALITY-ORIENTED EXPERTS

To provide a sound foundation for MCoT, we first introduce Functionality-oriented eXperts (FoXperts), an expert-parallel architecture that assigns experts based on function. In this way, FoXperts disentangle the potential conflicts in current mainstream modality-oriented designs. We find that predominant modality-oriented architectures (Liang et al., 2024; Shi et al., 2024) assign experts solely based on modality and handle both visual understanding and generation tasks within a single shared visual expert. However, visual understanding and visual generation pursue fundamentally different objectives: the former aligns image features with text for task performance, optimized by objectives such as Eq. 2, while the latter focuses on noise prediction for denoising, optimized by Eq. 4. Requiring a single shared expert to address both inevitably can lead to functional domain conflicts. Additionally, our functionality-oriented design motivation is also supported by prior work (Zhang et al., 2023b), which demonstrates emergent modularity in pre-trained Transformers, where neurons naturally cluster into functionally specialized modules, consistent with our approach.

In detail, FoXperts retain a unified text branch *Linguistic Expert*, since text understanding and generation share identical optimization objectives. For vision, however, we introduce two separate branches: a *Semantic Vision Expert* for visual understanding and a *Generative Vision Expert* for visual generation, each optimized with distinct objectives. The Linguistic Expert is essentially identical to Qwen2, initialized from the pre-trained Qwen2-0.5B model to preserve text-related capabilities. The initialization of the two visual experts, along with the detailed training strategy, is provided in Appendix C. All experts consist of $L$ layers with structurally consistent designs. Each expert maintains independent weights for all non-embedding parameters, including projection matrices in the attention module, feed-forward networks, and layer normalization, while sharing the multimodal routing module and the global multimodal attention module.

## 3.4 FORWARD PROCESS

Denote the input as $x = x_T \oplus x_C \oplus x_N$, where $x_T, x_C, x_N$ correspond to text tokens $x_{text}$, clean image tokens $x_{clean}$, and noise tokens $x_{noised}$, respectively. The forward pass of a single **FoXperts** layer can then be expressed as:

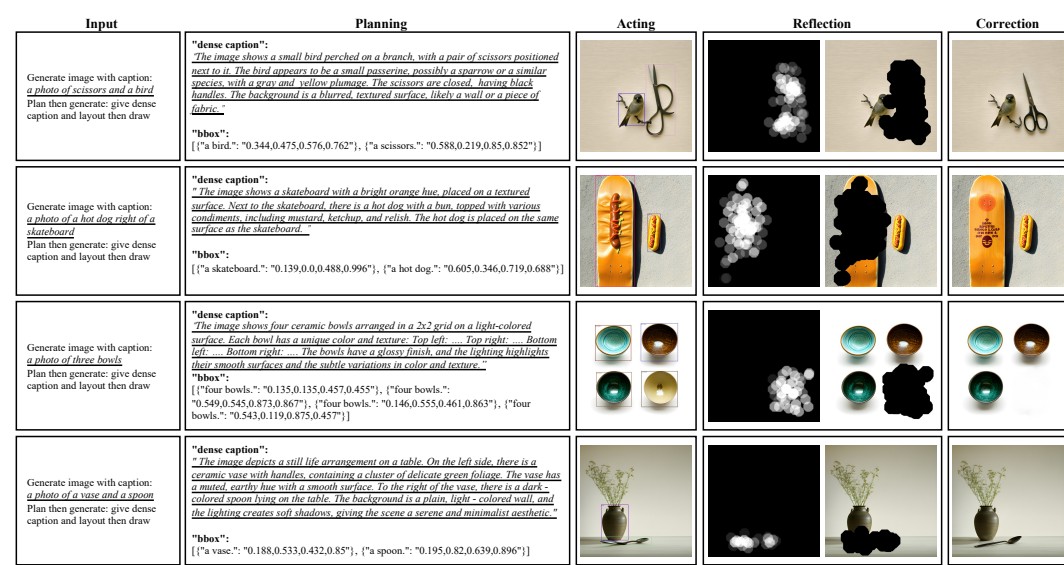

Figure 3: Examples of the full MCoT process. Each row shows a type of image defect that can be corrected through the reflection and correction steps. From top to bottom: Structural Incompleteness, Object Entanglement, Object Redundancy, and Object Distortion.

$$
\begin{aligned}
\hat{\boldsymbol{x}}_i &= W_i(Router(LN(\boldsymbol{x}))), \quad i \in \{T, C, N\} \\
\hat{\boldsymbol{x}}_i^{q,k,v} &= W_i^{Q,K,V}(\hat{\boldsymbol{x}}_i), \quad i \in \{T, C, N\} \\
\hat{\boldsymbol{x}}^{rep} &= \hat{\boldsymbol{x}}_T^{rep} \oplus \hat{\boldsymbol{x}}_C^{rep} \oplus \hat{\boldsymbol{x}}_N^{rep}, \quad rep \in \{q, k, v\} \\
\hat{\boldsymbol{x}} &= Attn(\hat{\boldsymbol{x}}^q, \hat{\boldsymbol{x}}^k, \hat{\boldsymbol{x}}^v) + \boldsymbol{x},
\end{aligned}
\tag{5}
$$

where $\oplus$ denotes concatenation; $T, C, N$ indicate Linguistic, Semantic Vision, and Generative Vision Experts, respectively. $Attn$ denotes the Multimodal Attention module. While expert parameters are decoupled, outputs interact and align at each layer through multimodal attention like (Liang et al., 2024; Shi et al., 2024). Locally, language tokens use causal attention, whereas vision tokens (clean and noisy) adopt bidirectional attention. Globally, all tokens follow a causal sequence, ensuring efficient loss and gradient computation without future information leakage.

The corresponding FFN module is simplified as:

$$
\begin{aligned}
\hat{\boldsymbol{x}}_i &= FFN_i(Router(LN(\boldsymbol{x}))), \quad i \in \{T, C, N\} \\
\hat{\boldsymbol{x}} &= \hat{\boldsymbol{x}}_T \oplus \hat{\boldsymbol{x}}_C \oplus \hat{\boldsymbol{x}}_N.
\end{aligned}
\tag{6}
$$

### 3.5 MULTIMODAL CHAIN OF THOUGHT FOR ENHANCED IMAGE GENERATION

Inspired by the human artistic workflow, where an artist first sketches object positions and outlines details, then reflects on and adjusts specific regions to achieve perfection, we propose the MCoT approach for enhanced image generation. Specially, MCoT is defined as a sequence of explicit key reasoning steps–**Planning, Acting, Reflection, and Correction**, following the human-defined key steps manner in previous works (Zhou et al., 2023; Jiang et al., 2024; Mitra et al., 2024; Xu et al., 2024; Guo et al., 2025b; Zhao et al., 2025).

#### 3.5.1 PLANNING AND ACTING

**Planning.** Emulating an artist sketching object positions and outlining details before painting, the planning step—as illustrated in the full process results in Figure 3—consists of two components: detailed caption planning and layout box planning. Detailed caption planning drafts a more comprehensive and precise image caption without distorting the meaning of the input prompt. Layout box planning assigns each object in the input prompt to a reasonable relative position in a bounding-box manner, learned from large-scale image layouts.

**Acting.** In the acting step, FoX generates images guided by the dense captions and layout boxes produced during the planning stage. We observe that using dense captions enhances image fidelity, consistent with prior works (Hao et al., 2023; Jo et al., 2025) indicating that longer and more detailed prompts often yield better image quality. Furthermore, the layout boxes enable our model to place objects accurately within the specified areas. Additional examples demonstrating improvements in fidelity and adherence to layout are provided in Figure 5 in Appendix B.

### 3.5.2 REFLECTION AND CORRECTION

**Reflection.** Despite the planning and acting steps improving image generation, the model may still fail to render all parts of an image perfectly in a single attempt. Four main types of image defects that can be corrected through the reflection and correction steps are illustrated in Figure 3. In the reflection step, the model uses the generated image and the input prompt as conditions to identify regions with defects, such as low aesthetic quality or misalignment with the prompt. This process generates an artifact heatmap, where higher confidence scores indicate areas requiring more substantial corrections.

**Correction.** In the correction step, our model integrates the artifact reflection map and planning rationale into the generation process, as shown in Figure 2. Using this information, the model performs targeted inpainting to refine masked regions based on the artifact heatmap. Additional correction examples are provided in Figure 3.

### 3.6 MULTI-TASK JOINT TRAINING PARADIGM

We develop a multi-task joint training paradigm that equips the model with all capabilities in a disentangled manner, overcoming the challenge of collecting consistent multi-step data tuples. It would be unavoidable if we adopt straightforward end-to-end training with fully supervised data across the entire process, as discussed in the introduction. Specifically, we decouple the end-to-end MCoT training process into multiple tasks based on the MCoT steps: Planning and Acting Task, Reflection Task, and Correction Task. This approach allows us to train the model using more readily available data for each sub-task, avoiding the drawback of forcing the model to generate erroneous images after planning in end-to-end MCoT training.

For Planning and Acting Task, our model is trained with quadruples consisting of {input prompt, detailed caption, layout box, image}, expanded from prompt-image data pairs, where the model takes the input prompt and generates the detailed caption, layout box, and image. For Reflection Task, the model is trained to take the first generated image and input prompt to generate an artifact map, with artifact map labels produced from existing datasets and manually annotated data. For Correction Task, the model is trained in a standard inpainting style, refining masked regions based on all previous results such as detailed caption planning and the first generated image. Additional training details and data construction methods are provided in Appendix C and D.

## 4 EXPERIMENTS

### 4.1 IMPLEMENTAL DETAILS

We employ AdamW (Loshchilov & Hutter, 2017) as the optimizer with parameter setting of $\beta_1 = 0.9, \beta_2 = 0.999$, and a weight decay of 0.02. The learning rate is configured to a constant value of $5 \times 10^{-5}$, incorporating a linear warm-up phase over 10,000 steps. The training utilized DeepSpeed's ZeRO-2 (Rajbhandari et al., 2020) optimization.

### 4.2 MAIN RESULTS FOR ENHANCED IMAGE GENERATION

**GenEval Benchmarks.** We evaluate on GenEval (Ghosh et al., 2024), a compositional image benchmark for assessing generative models under complex conditions. As shown in Table 1, FoX achieves an overall score of 0.77, outperforming all models in its category. Across individual tasks, it delivers superior or comparable performance to both unimodal and unified generative models, despite having only 1.3 billion parameters. Furthermore, our FoX outperforms larger unimodal T2I models on GenEval, delivering a 9% gain over SD3 and 10% over DALL·E 3; notably, it achieves 20% and 19% improvements on the challenging Attribute Binding metric, respectively. More qualitative comparisons with current models including SD3 are provided in Appendix B.

Table 1: **Comparison of enhanced image generation quality on GenEval.** "Uni." refers to unimodal generative models that operate exclusively on images, while "Multi." indicates multimodal generative models that are capable of generating both images and text.

| Model | Params | Type | Overall↑ | Single Obj. | Two Obj. | Counting | Colors | Position | Attr. Binding |
|---|---|---|---|---|---|---|---|---|---|
| SD v1.5 Rombach et al. (2022) | 1B | Uni. | 0.43 | 0.97 | 0.38 | 0.35 | 0.76 | 0.04 | 0.06 |
| SD v2.1 Rombach et al. (2022) | 1.3B | Uni. | 0.50 | 0.98 | 0.51 | 0.44 | 0.85 | 0.07 | 0.17 |
| SD-XL Podell et al. (2024) | 3.4B | Uni. | 0.55 | 0.98 | 0.74 | 0.39 | 0.85 | 0.15 | 0.23 |
| SD 3 Esser et al. (2024a) | 12.7B | Uni. | 0.68 | 0.98 | 0.84 | 0.66 | 0.74 | 0.40 | 0.43 |
| DALL-E 2 Ramesh et al. (2022) | 4.5B | Uni. | 0.52 | 0.94 | 0.66 | 0.49 | 0.77 | 0.10 | 0.19 |
| DALL-E 3 Betker et al. (2023a) | – | Uni. | 0.67 | 0.96 | 0.87 | 0.47 | 0.83 | 0.43 | 0.45 |
| IF-XL DeepFloyd (2023) | 10.1B | Uni. | 0.61 | 0.97 | 0.74 | 0.66 | 0.81 | 0.13 | 0.35 |
| Chameleon Team (2024a) | 34B | Multi. | 0.39 | – | – | – | – | – | – |
| Transfusion Zhou et al. (2024a) | 7.3B | Multi. | 0.63 | – | – | – | – | – | – |
| LWM Liu et al. (2024a) | 7B | Multi. | 0.47 | 0.93 | 0.41 | 0.46 | 0.79 | 0.09 | 0.15 |
| SEED-X Ge et al. (2024a) | 17B | Multi. | 0.49 | 0.97 | 0.58 | 0.26 | 0.80 | 0.19 | 0.14 |
| Show-o Xie et al. (2024a) | 1.3B | Multi. | 0.53 | 0.95 | 0.52 | 0.49 | 0.82 | 0.11 | 0.28 |
| Janus Wu et al. (2024a) | 1.3B | Multi. | 0.61 | 0.97 | 0.68 | 0.30 | 0.84 | 0.46 | 0.42 |
| JanusFlow Ma et al. (2024) | 1.3B | Multi. | 0.63 | 0.97 | 0.59 | 0.45 | 0.83 | 0.53 | 0.42 |
| **FoX (Ours)** | 1.3B | Multi. | **0.77** | **0.99** | **0.86** | **0.71** | 0.82 | **0.60** | **0.64** |

Table 2: **Comparison of enhanced image generation quality on T2I-CompBench.** This evaluation demonstrates performance generalization, further supporting the robustness and effectiveness of our approach across diverse scenarios.

| Model | Params | Color↑ | Shape↑ | Texture↑ | Spatial↑ | Non-Spatial↑ | Complex↑ |
|---|---|---|---|---|---|---|---|
| SD v1.5 Rombach et al. (2022) | 1B | 37.65 | 35.76 | 41.56 | 12.46 | 30.79 | 30.80 |
| SD-XL Podell et al. (2024) | 3.4B | 63.69 | 54.08 | 56.37 | 20.32 | 31.10 | 40.91 |
| SD3 Esser et al. (2024a) | 12.7B | 81.32 | 58.85 | 73.34 | 32.00 | 31.40 | 37.71 |
| GORS Huang et al. (2023) | - | 66.03 | 47.85 | 62.87 | 18.15 | 31.93 | 33.28 |
| PixArt-$\alpha$ Chen et al. (2023) | 0.6B | 68.86 | 55.82 | 70.44 | 20.82 | 31.79 | 41.17 |
| DALL-E 2 Ramesh et al. (2022) | 4.5B | 57.50 | 54.64 | 63.74 | 12.83 | 30.43 | 36.96 |
| T2I-R1 Jiang et al. (2025) | 7B | 81.30 | 58.52 | 72.43 | 33.78 | 30.90 | 39.93 |
| GoT-R1 Duan et al. (2025) | 7B | 81.39 | 55.49 | 73.39 | 33.06 | 31.69 | 39.44 |
| EMU3 Wang et al. (2024b) | 7B | 75.44 | 57.06 | 71.64 | - | - | - |
| Janus-Pro Chen et al. (2025) | 7B | 63.59 | 35.28 | 49.36 | 20.61 | 30.85 | 35.59 |
| Show-o Xie et al. (2024b) | 1.3B | 56 | 41 | 46 | 20 | 30 | 29 |
| **FoX (Ours)** | 1.3B | **82.37** | **59.81** | **74.21** | **35.71** | **34.19** | **42.78** |

**T2I-CompBench Benchmarks.** We evaluate FoX on T2I-CompBench (Huang et al., 2023) to assess generalization and robustness. As shown in Table 2, FoX achieves state-of-the-art scores in Color (82.37), Spatial (35.71), Non-Spatial (34.19), and Complex (42.78), demonstrating superior performance in color accuracy, spatial arrangement, semantic consistency, and compositional complexity for complex image generation. Notably, larger models including Janus-Pro and Emu3 still trail our performance. Moreover, other CoT-based methods (T2I-R1 and GoT-R1) underperform our approach on T2I-CompBench, even with their larger 7B architectures.

**MS-COCO Benchmarks.** We evaluate FoX on MS-COCO (Lin et al., 2014) to demonstrate foundational image generation capabilities. As shown in Table 3, FoX achieves FID (Heusel et al., 2017) 7.24, outperforming DALL-E 2 (10.39) and NExT-GPT (10.07), with a CLIP score (Radford et al., 2021) of 26.8 comparable to Transfusion (25.5) despite fewer parameters. FoX also attains the highest CIDEr (Vedantam et al., 2015) score of 126.5, indicating strong text-image alignment and multimodal generation effectiveness.

## 4.3 ADDITIONAL RESULTS FOR IMAGE UNDERSTANDING

As shown in Table 4, FoX achieves strong results across multiple image understanding benchmarks. It attains an MME-P (Fu et al., 2024) score of 1339.7, outperforming comparable unimodal models and closely approaching top performers with fewer parameters. In MMBench (Liu et al., 2024e), FoX scores 73.6, surpassing competitors such as Janus and LLaVA-v1.5. Moreover, with a VQA-v2 p (Goyal et al., 2017) score of 79.4, FoX ranks among the leading models in visual question answering. These results demonstrate that FoX is highly competitive across both text-only unimodal and multimodal generative frameworks in image understanding tasks. To comprehensively evaluate our FoX's multimodal understanding, we further assess it on three additional representative benchmarks (MMMU (Yue et al., 2024), MM-Vet (Yu et al., 2023), TextVQA (Yang et al., 2021)), with results in Table 10. The results indicate that FoX's multimodal understanding capabilities are comparable to current similar-scale models.

Table 3: **Comparison of foundational image generation on MS-COCO.** * indicates that CIDEr is calculated based on 30K randomly sampled data from validation set, rather than Karpathy test split set.

| Model | Params | FID↓ | CLIP Score ↑ | CIDEr↑ |
|---|---|---|---|---|
| *Uni-modal Generative Model* | | | | |
| SD v1.5 Rombach et al. (2022) | 1B | 9.93 | 30.2 | – |
| SD-XL Podell et al. (2024) | 3.4B | – | 31.0 | – |
| DALL-E 2 Ramesh et al. (2022) | 4.5B | 10.39 | 31.4 | – |
| DALL-E 3 Betker et al. (2023a) | – | – | 32.0 | – |
| IF-XL DeepFloyd (2023) | 10.1B | 6.66 | – | – |
| Emu2-GEN Sun et al. (2024a) | 37B | – | 29.7 | – |
| *Multi-modal Generative Model* | | | | |
| DREAMLLM Dong et al. (2024) | 7B | – | – | 115.4 |
| Chameleon Team (2024b) | 7B | 26.74 | 24.3 | 120.2 |
| Transfusion Zhou et al. (2024b) | 7.3B | 6.78 | 25.5 | 32.0* |
| SEED-LLaMA Ge et al. (2023) | 8B | – | – | 123.6 |
| MetaMorph Tong et al. (2024) | 8B | 11.8 | – | – |
| Emu 3 Wang et al. (2024b) | 8B | 12.8 | 31.3 | – |
| LLamaFusion Shi et al. (2024) | 8B | 8.61 | 24.4 | 38.4* |
| LLaVAFusion Shi et al. (2024) | 8B | 8.28 | 24.7 | – |
| NExT-GPT Wu et al. (2024b) | 13B | 10.07 | – | 124.9 |
| Show-o Xie et al. (2024a) | 1.3B | 9.24 | – | – |
| Janus Wu et al. (2024a) | 1.3B | 8.5 | – | – |
| **FoX (Ours)** | 1.3B | 7.24 | 26.8 | 126.5 |

Table 4: **Comparison results of foundational image understanding performance.**

| Model | Params | MME-P↑ | MMBench↑ | VQA-v2↑ |
|---|---|---|---|---|
| *Uni-modal Generative Model* | | | | |
| MobileVLM Chu et al. (2023) | 2.7B | 1288.9 | 59.6 | – |
| LLaVA-Phi Zhu et al. (2024) | 2.7B | 1335.1 | 59.8 | 71.4 |
| LLaVA Liu et al. (2024c) | 7B | 809.6 | 38.7 | – |
| LLaVA-v1.5 Liu et al. (2024b) | 7B | 1510.7 | 64.3 | 78.5 |
| Qwen-VL-Chat Bai et al. (2023) | 7B | 1487.5 | 60.6 | 78.2 |
| IDEFICS-9B Laurençon et al. (2023) | 8B | – | 48.2 | 50.9 |
| Emu3-Chat Wang et al. (2024b) | 8B | – | 58.5 | 75.1 |
| InstructBLIP Dai et al. (2023) | 13B | 1212.8 | – | – |
| LLaVA-v1.5-Phi-1.5 Xie et al. (2024a) | 1.3B | 1128.0 | – | 75.3 |
| MobileVLM Chu et al. (2023) | 1.4B | 1196.2 | 53.2 | – |
| MobileVLM-V2 Chu et al. (2024) | 1.4B | 1302.8 | 57.7 | – |
| *Multi-modal Generative Model* | | | | |
| LWM Liu et al. (2024a) | 7B | – | – | 55.8 |
| VILA-U Wu et al. (2024c) | 7B | 1401.8 | – | 79.4 |
| LaVIT Jin et al. (2024) | 7B | – | – | 66.0 |
| ChameleonTeam (2024a) | 7B | – | 35.7 | – |
| Emu Sun et al. (2024b) | 13B | – | – | 52.0 |
| NExT-GPT Wu et al. (2024b) | 13B | – | – | 66.7 |
| LLaVAFusion Shi et al. (2024) | – | – | 72.1 | – |
| Gemini-Nano-1 Team (2023) | 1.8B | – | – | 62.7 |
| Show-o Xie et al. (2024a) | 1.3B | 948.4 | – | 59.3 |
| JanusFlow Ma et al. (2024) | 1.3B | 1333.1 | 74.9 | 79.8 |
| Janus Wu et al. (2024a) | 1.3B | 1338.0 | 69.4 | 77.3 |
| **FoX (Ours)** | 1.3B | 1339.7 | 73.6 | 79.4 |

Table 5: **Ablation results on GenEval Benchmark for validating the effectiveness of MCoT.**

| Setting | Single Obj. | Two Obj. | Counting | Colors | Position | Attr. Binding | Overall↑ |
|---|---|---|---|---|---|---|---|
| T2I Gen. Twice | 0.97 | 0.80 | 0.58 | 0.78 | 0.40 | 0.47 | 0.67 |
| MCoT Planning & Acting Only | 0.98 | 0.84 | 0.66 | 0.81 | 0.55 | 0.58 | 0.73 |
| MCoT Full Process | 0.99 | 0.86 | 0.71 | 0.82 | 0.60 | 0.64 | 0.77 |

Table 6: **Ablation results on T2I-CompBench for validating the effectiveness of MCoT.**

| Setting | Color | Shape | Texture | Spatial | Non-Spatial | Complex | Overall↑ |
|---|---|---|---|---|---|---|---|
| T2I Gen. Twice | 65.15 | 51.36 | 64.05 | 13.42 | 26.89 | 32.72 | 42.26 |
| MCoT Planning&Acting Only | 76.77 | 56.70 | 71.08 | 28.36 | 31.28 | 39.55 | 50.62 |
| MCoT Full Process | 82.37 | 59.81 | 74.21 | 35.71 | 34.19 | 42.78 | 54.85 |

### 4.4 ADDITIONAL RESULTS FOR QUALITATIVE ANALYSIS

Figure 1 shows that FoX with MCoT overcomes T2I limitations like concept confusion. Figure 3 presents the full MCoT process: planning adds details without changing the prompt, acting follows layout accurately, and reflection and correction steps fix defects such as structural incompleteness. Figure 4 compares FoX with other models, highlighting its superior image generation quality. Figure 4 and 7 demonstrate that FoX maintains strong prompt alignment and high generation fidelity, achieving superior quality among widely used models including SD3, SD-XL, and JanusFlow.

### 4.5 ABLATION STUDY

**Ablation of MCoT.** We assess MCoT's effectiveness on GenEval and T2I-CompBench as shown in Tables 5 and 6. To ensure fairness, the baseline model is trained for additional epochs using the pre-training paradigm on the same checkpoint, eliminating biases from extra MCoT training steps. Since the full MCoT process involves two image generations, the baseline performs T2I generation twice at test time, selecting the best result as reference. Results show that FoX already outperforms the baseline with only planning and acting steps, and adding reflection and correction further improves performance, fully validating MCoT's effectiveness. Reflection-Correction steps yield notable gains even when applied to the outputs of Planning-Acting steps that already deliver corrected results, validating the necessity of Reflection-Correction steps. We provide more ablation results to better isolate and validate the effectiveness of Reflection-Correction in Appendix A.

**Ablation of FoXperts.** We conducted pre-training experiments under identical settings across three variants: dense design, modality-oriented design, and our functionality-oriented FoX. As shown in Table 7 on MS-COCO, FoX outperforms both dense and modality-oriented variants: the dense model achieves an FID of 11.3 and a CIDEr of 116.2, the modality-oriented model achieves an FID of 9.56

Table 7: **Ablation of FoXperts.**

| Model | CIDEr↑ | FID↓ |
|---|---|---|
| Dense | 116.2 | 11.3 |
| Modality-Oriented | 121.1 | 9.56 |
| FoX (Ours) | 126.5 | 7.24 |

and a CIDEr of 121.1, while FoX attains an FID of 7.24 and a CIDEr of 126.5. These results demonstrate that FoX enhances both image understanding and generation through a functionality-disentangled design for visual experts, validating that our architecture alleviates conflicts inherent in a single shared visual expert. More detailed experimental description and discussion of functionality conflicts are provided in Appendix E.

**Additional Results for Visual Quality.** To verify that our approach improves text-to-image alignment without compromising visual quality, we evaluated 300 prompts from GenEval using Aesthetic Score (LAION-AI, 2023) and Human Preference Score v2 (HPS v2) (Wu et al., 2023). Image triplets were generated for T2I Generation, Planning & Acting, and the Full MCoT Process. As shown in Table 8, Aesthetic Scores remain stable despite improvements in text-to-image alignment, while HPS v2 win rates demonstrate consistent human preference for Planning & Acting and MCoT outputs over baseline T2I generations, reflecting enhanced alignment, composition, and detail beyond what Aesthetic Scores capture.

Table 8: **Results of visual quality on Aesthetic Benchmarks.**

| Aesthetic Score | |
|---|---|
| **Setting** | **Score** |
| T2I Gen. | 5.98 |
| Planning & Acting | 6.02 |
| Full MCoT | 6.04 |
| **HPS v2 Score (Win Rate)** | |
| **Setting** | **Score** |
| Planning & Acting vs. T2I | 57.8 |
| Full MCoT vs. T2I | 62.3 |

## 5 CONCLUSIONS

We introduce **FoX**, a unified generative model featuring the FoXperts architecture and an MCoT approach to address the limitations of direct text-to-image generation in complex scenarios. FoXperts disentangle expert functionalities, mitigating conflicts in modality-oriented designs, while MCoT structures image generation into planning, acting, reflection, and correction. A multi-task joint training paradigm ensures effective learning for each step. Experiments across multiple benchmarks validate the enhanced image generation capability of FoX. These results demonstrate that integrating functional expert design with stepwise multimodal reasoning substantially improves complex image synthesis.

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

# A  ADDITIONAL QUANTITATIVE RESULTS

**Additional ablation study for Reflection-Correction.** As shown in Tables 5 and 6, the Reflection-Correction steps yield notable gains even when applied to the outputs of Planning-Acting steps that already deliver corrected results. This thus underscores the mechanism's efficacy despite the high bar for further improvement. To better isolate and validate the effectiveness of Reflection-Correction, we conduct an additional ablation study on T2I-CompBench, where we replace Planning-Acting with direct T2I generation followed by Reflection-Correction. Table 9 shows that Reflection-Correction yields a 15.6% Overall score gain (42.26→48.86) over the T2I Gen. Twice setting. Particularly, the steps boost the Spatial metric by a striking 90.2% (13.42→25.72), validating their efficacy for challenging spatial constraints.

Table 9: **Ablation results of Reflection and Correction on T2I-CompBench.**

| Setting | Color | Shape | Texture | Spatial | Non-Spatial | Complex | Overall↑ |
|---|---|---|---|---|---|---|---|
| T2I Gen. Twice | 65.15 | 51.36 | 64.05 | 13.42 | 26.89 | 32.72 | 42.26 |
| T2I Gen. + Reflection & Correction | 74.13 | 55.27 | 69.86 | 25.72 | 30.21 | 37.95 | 48.86 |

Table 10: **Additional Results on Multimodal Understanding Benchmarks.**

| Model | Params | MMMU | MM-Vet | TextVQA |
|---|---|---|---|---|
| TokenFlow-XL | 13B | 38.7 | 40.7 | - |
| SEED-X | 13B | 35.6 | 43.0 | - |
| Emu3-Chat | 8B | 31.6 | 37.2 | 64.7 |
| Qwen2-VL | 7B | 54.1 | 62.0 | 84.3 |
| ILLUME | 7B | 38.2 | 37.0 | - |
| LLaVA-v1.5 | 7B | 35.4 | 31.1 | - |
| VILA-U | 7B | - | 33.5 | 60.8 |
| Chameleon | 7B | 22.4 | 8.3 | - |
| BLIP-3 | 4B | 41.1 | - | - |
| InternVL2 | 1.8B | 34.3 | 44.6 | - |
| Gemini-Nano-1 | 1.8B | 26.3 | - | 62.5 |
| LLaVA-v1.5-Phi-1.5 | 1.3B | 30.7 | - | - |
| Show-o | 1.3B | 25.1 | - | - |
| Janus | 1.3B | 30.5 | 34.3 | - |
| JanusFlow | 1.3B | 29.3 | 30.9 | 55.5 |
| FoX (Ours) | 1.3B | 31.3 | 33.7 | 57.2 |

Table 11: **Scaling-up experiments on GenEval.** Scaling up improves all metrics consistently, supporting the method's scalability.

| Model | Params | Overall↑ | Single Obj. | Two Obj. | Counting | Colors | Position | Attr. Binding |
|---|---|---|---|---|---|---|---|---|
| FoX (Original) | 1.3B | 0.77 | 0.99 | 0.86 | 0.71 | 0.82 | 0.60 | 0.64 |
| FoX (Scale-up) | 4.2B | 0.82 | 0.99 | 0.89 | 0.76 | 0.87 | 0.68 | 0.70 |

**Scaling-up results.** Table 11 presents our scale-up results on GenEval using the larger Qwen2-1.5B backbone. Scaling the backbone from Qwen2-0.5B to Qwen2-1.5B yields consistent improvements across all metrics, which supports the scalability of our method. In particular, the Position score increases from 0.60 to 0.68 and the Attribute Binding score from 0.64 to 0.70, showing clear gains on challenging aspects. Moreover, our scaled-up FoX (4.2B) achieves overall performance comparable to the reported results of BAGEL (7B) on GenEval. Notably, it outperforms BAGEL on challenging metrics in complex image generation, including Position (0.68 vs. 0.64) and Attr. Binding (0.70 vs. 0.63), despite utilizing merely 60% of the latter's parameter count (4.2B vs. 7B).

**Additional results for image generation.** We further evaluate our model on the WISE benchmark, a representative and widely adopted reasoning-related image generation benchmark. The results in Table 12 demonstrate that our model achieves performance comparable to other models of similar or even larger scales, further validating the broad applicability and effectiveness of our MCoT framework.

Table 12: **Additional Results of Image Generation on WISE Benchmark.**

| Model | Params | Cultural | Time | Space | Biology | Physics | Chemistry | Overall |
|---|---|---|---|---|---|---|---|---|
| SD-3 | 12.7B | 0.42 | 0.44 | 0.48 | 0.39 | 0.47 | 0.29 | 0.42 |
| T2I-R1 | 7B | 0.56 | 0.55 | 0.63 | 0.54 | 0.55 | 0.30 | 0.54 |
| Emu3 | 7B | 0.34 | 0.45 | 0.48 | 0.41 | 0.45 | 0.27 | 0.39 |
| Janus-Pro | 7B | 0.30 | 0.37 | 0.49 | 0.36 | 0.42 | 0.26 | 0.35 |
| VILA-U | 7B | 0.26 | 0.33 | 0.37 | 0.35 | 0.39 | 0.23 | 0.31 |
| SD-XL-base-0.9 | 3.4B | 0.43 | 0.48 | 0.47 | 0.44 | 0.45 | 0.27 | 0.43 |
| Show-o | 1.3B | 0.28 | 0.36 | 0.40 | 0.23 | 0.33 | 0.22 | 0.30 |
| Janus | 1.3B | 0.16 | 0.26 | 0.35 | 0.28 | 0.30 | 0.14 | 0.23 |
| JanusFlow | 1.3B | 0.13 | 0.26 | 0.28 | 0.20 | 0.19 | 0.11 | 0.18 |
| Janus-Pro | 1B | 0.20 | 0.28 | 0.45 | 0.24 | 0.32 | 0.16 | 0.26 |
| FoX (Ours) | 1.3B | 0.46 | 0.52 | 0.65 | 0.41 | 0.48 | 0.27 | 0.47 |

Table 13: **Ablation Results of Reflection-Correction Cycles on GenEval.**

| Setting | Overall↑ | Single Obj. | Two Obj. | Counting | Colors | Position | Attr. Binding |
|---|---|---|---|---|---|---|---|
| MCoT with one reflection–correction cycle | 0.77 | 0.99 | 0.86 | 0.71 | 0.82 | 0.60 | 0.64 |
| MCoT with two reflection–correction cycles | 0.78 | 0.99 | 0.88 | 0.73 | 0.82 | 0.61 | 0.65 |

**Ablation results of Reflection-Correction cycles in MCoT.** Our reflection and correction steps can be performed multiple times. However, we find that a single reflection–correction cycle already provides a good balance between performance and efficiency. Executing multiple cycles yields only marginal gains while incurring additional inference cost. In Table 13, we present a comparison on GenEval between running one versus two reflection–correction cycles. The results show that the improvements from a second cycle are relatively limited, with only minor changes across metrics. We also include qualitative examples for the two-cycle setting in Figure 8, where the first correction already meets most requirements and the second correction introduces only subtle adjustments.

# B    ADDITIONAL QUALITATIVE RESULTS

**Visual comparisons with current models.** Figure 4 demonstrates that even prominent T2I models such as SD3 suffer from issues like object defects in two-object generation tasks and spatial confusion in position-aware tasks, indicating that SD3 still struggles with complex image generation scenarios. In contrast, our model maintains strong prompt alignment and high generation fidelity, achieving the superior performance among these widely used models. We further evaluate the practical performance of SD3 on challenging examples in Figure 1 containing three or four objects—a scenario commonly encountered in practice—and compare the results with ours. As shown in Figure 7, the issues for SD3 are even more pronounced, confirming that complex image generation tasks remain challenging. Importantly, the images produced by our model exhibit strong adherence to input prompts and high fidelity, demonstrating that our MCoT framework effectively enhances the model's ability to solve these tasks.

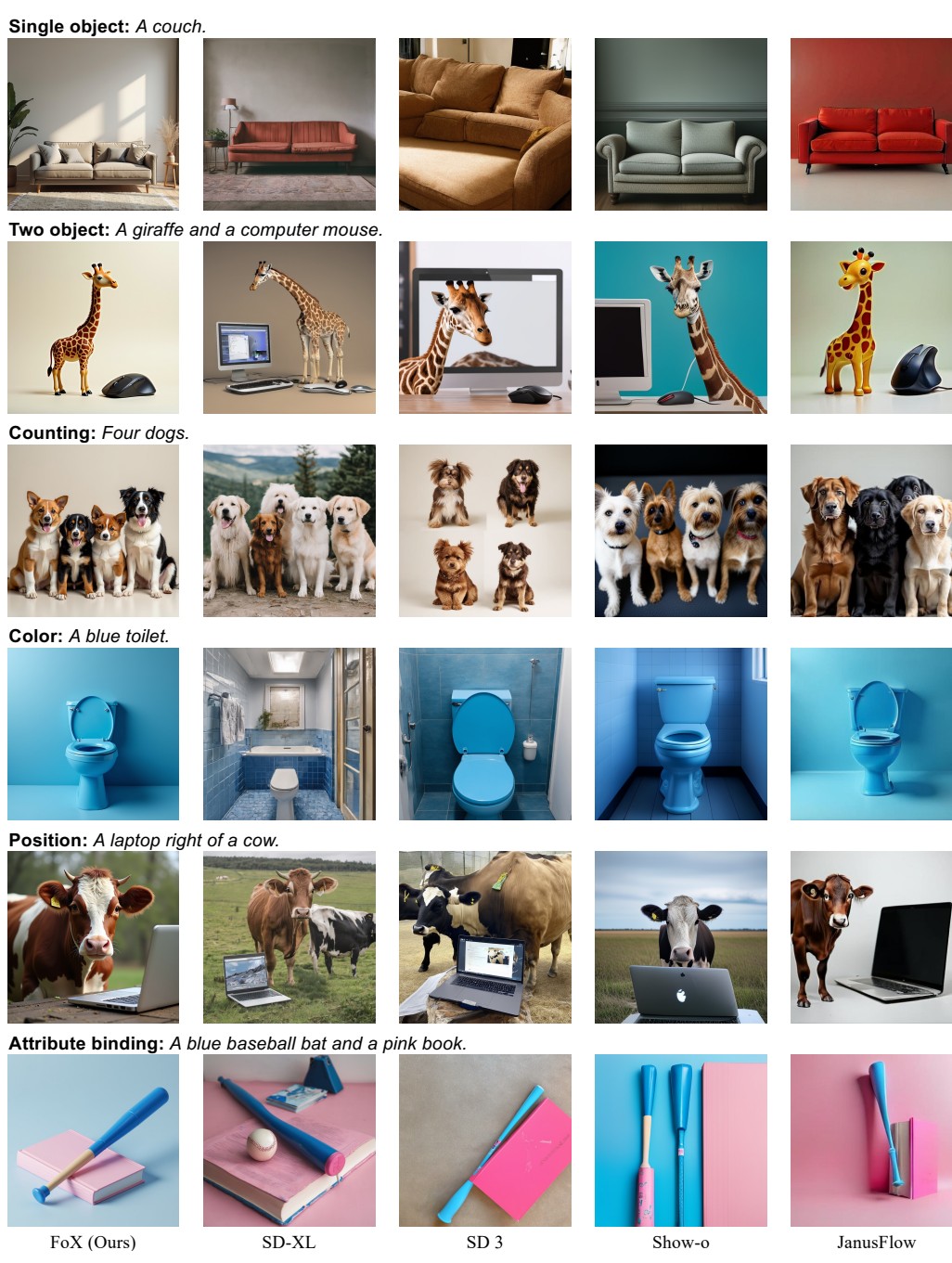

**Single object:** *A couch.*

**Two object:** *A giraffe and a computer mouse.*

**Counting:** *Four dogs.*

**Color:** *A blue toilet.*

**Position:** *A laptop right of a cow.*

**Attribute binding:** *A blue baseball bat and a pink book.*

FoX (Ours)   SD-XL   SD 3   Show-o   JanusFlow

Figure 4: Qualitative comparisons with baseline models, including unimodal generative models such as SD-XL and SD 3, as well as multimodal generative models like Show-o and JanusFlow. These comparisons span the six evaluation categories of GenEval and are accompanied by the corresponding prompts to assess alignment between prompts and generated images.

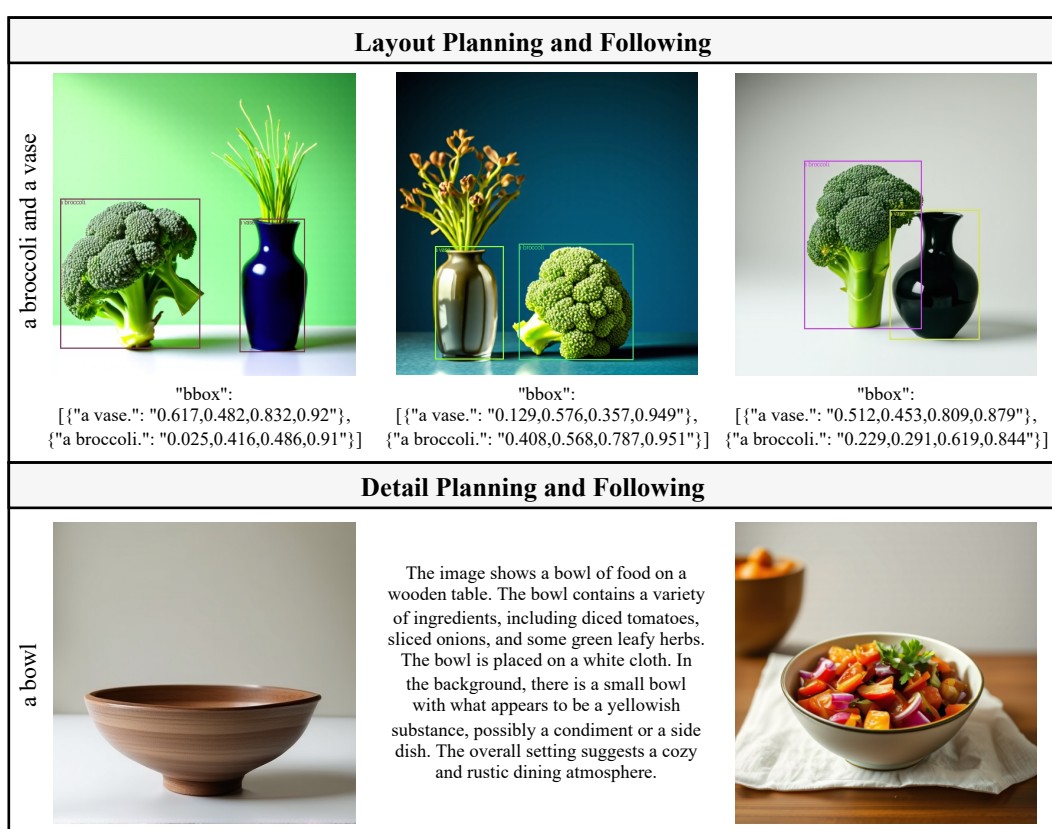

Figure 5: Layout planning enables our model to place objects accurately within the specified areas. Detail planning enhances image fidelity.

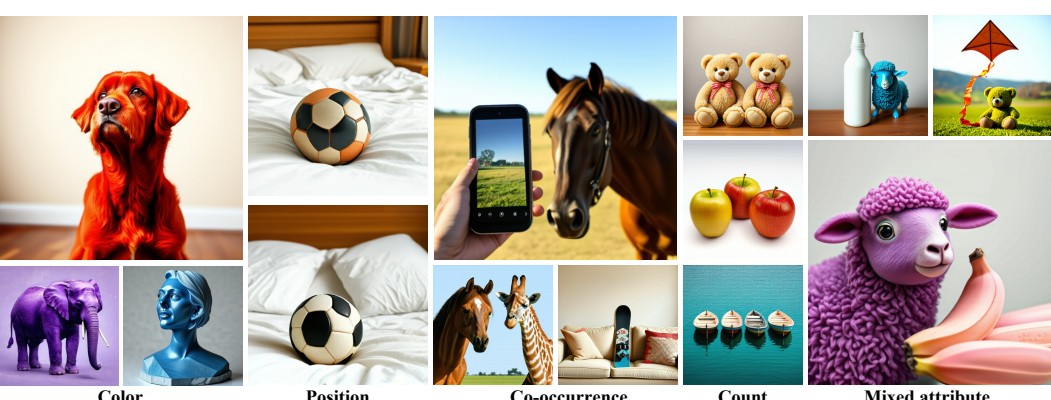

Figure 6: Additional examples generated by FoX under complex compositional prompts. Images are arranged left-to-right, top-to-bottom, with prompts (prefix "a photo of" omitted): a red dog, a purple elephant, a blue bust sculpture, a multi-colored soccer ball on a bed, a soccer ball on a bed, a cell phone and a horse, a horse and a giraffe, a couch and a snowboard, two teddy bears, three apples, four boats, a white bottle and a blue sheep, a green teddy bear below a brown kite, and a purple sheep to the left of a pink banana.

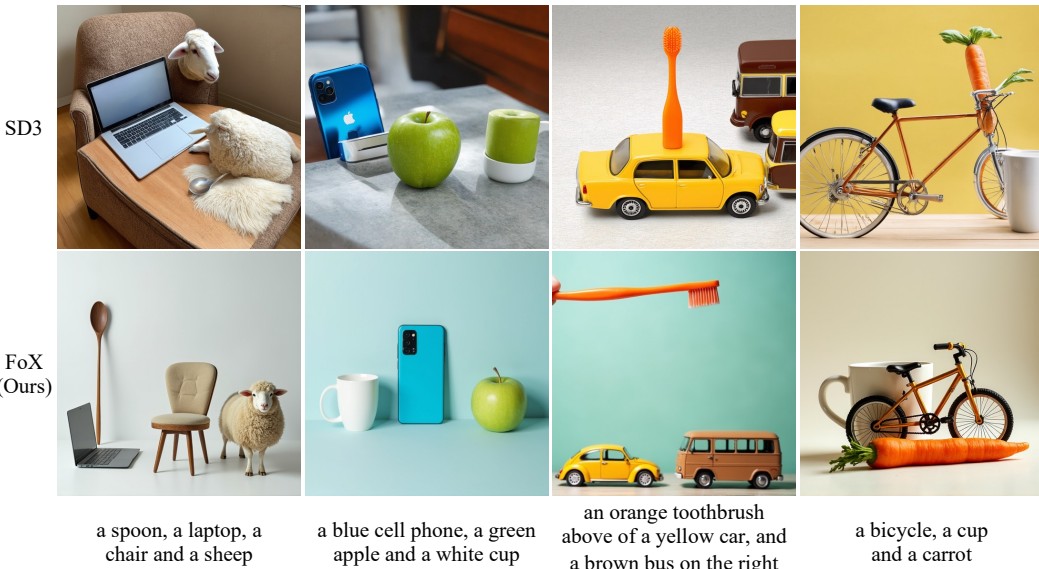

SD3

FoX
(Ours)

a spoon, a laptop, a
chair and a sheep

a blue cell phone, a green
apple and a white cup

an orange toothbrush
above of a yellow car, and
a brown bus on the right

a bicycle, a cup
and a carrot

Figure 7: Visual Comparisons with SD3 on More Complex Scenarios.

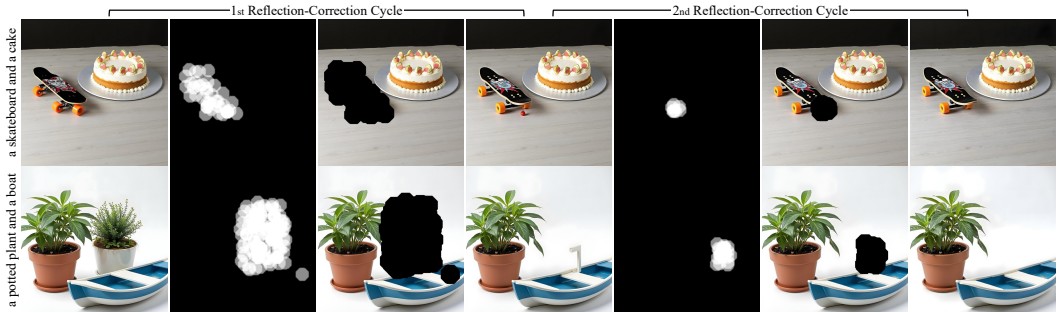

Figure 8: Visual results of Multiple Reflection-Correction Cycles.

## C TRAIN STRATEGY

### C.1 PRE-TRAINING FOR T2I AND I2T

We adopt a two-stage training approach to systematically integrate the text-to-image (T2I) and image-to-text (I2T) capabilities of the FoX model. For images with a resolution of $256 \times 256$ pixels, the batch size per GPU is set at 64, while for $512 \times 512$ pixel images, it is set at 16, leading to total batch sizes of 4096 and 1024, respectively.

**Stage I**: In this initial stage, we focus on optimizing the T2I functionality of FoX by training the model specifically for the T2I task. Both the Linguistic Expert and the Generative Visual Expert are trained, with their parameters initialized from the Qwen2-0.5B model. The image resolution for this phase is configured to $256 \times 256$ pixels.

**Stage II**: During this stage, FoX undergoes mixed training, encompassing both T2I and I2T tasks at a training task ratio of 8:1. The I2T tasks include image captioning and Visual Question Answering (VQA). In this phase, all experts are trainable, and the newly introduced Semantic Visual Expert is initialized using parameters from the Generative Visual Expert established in Stage I. The resolution of images at this stage is increased to $512 \times 512$.

**Initialization Strategies**: During pre-training, we first train the image-generation branch. When moving to the second-stage visual-understanding training, we initialize the Semantic Visual Expert using the weights of the Generative Visual Expert, as both belong to the image domain and are

Table 14: **Performance Comparison of Different Expert Initialization Strategies.**

| Setting | CIDEr↑ | FID↓ |
|---|---|---|
| From Generative Visual Expert | 124.6 | 7.83 |
| From Linguistic Expert | 122.1 | 7.97 |

then more closely aligned than the text branch, which might better facilitate image understanding. We did not choose to train the Semantic Visual Expert from scratch due to higher training costs and potentially unstable optimization, as similarly noted in Mono-internvl (Luo et al., 2025). And we compared two different initialization strategies for the Semantic Visual Expert under identical training settings. Evaluation results on MS-COCO, using CIDEr for image understanding and FID for image generation. As shown in the Table 14, our model initialized with the Generative Visual Expert demonstrates better performance.

### C.2 MULTIMODAL CHAIN OF THOUGHT TRAINING

We claim that executing the full end-to-end MCoT training presents challenges, requiring consistent data pairs across the entire process, and optimizing multiple processes simultaneously is difficult. Therefore, We designed a multi-task joint MCoT training framework that decouples the end-to-end MCoT training process into multiple training tasks. We divided the MCoT process into three training tasks: planning and acting, reflection, and correction. During training, the parameters of the Linguistic Expert, Generative Visual Expert, and Semantic Visual Expert are all trained. During joint training, the three tasks are alternately trained, with each task receiving equal training proportion, resulting in a 1:1:1 ratio among the three training tasks.

**Planning and Acting Task**: This task trains the model to perform both caption and layout planning. Analogous to how an artist sketches a composition and plans details for each part, our model is trained on quadruples of {input prompt, detailed caption, layout box, image}, expanded from prompt–image pairs. Given a prompt, the model generates a detailed caption, layout box, and final image. Dense captions are refined using Qwen-VL, while bounding boxes are extracted with Grounding-DINO-SAM by parsing object noun phrases in the captions. The open-source CC12M dataset was used, yielding a final dataset of 100K samples. The complete data construction pipeline is illustrated in Figure 9.

**Reflection Task**: This task equips the model with self-reflection capabilities, enabling it to detect regional defects and misalignments in the preliminary generated image relative to the caption. Regions requiring correction are represented through artifact map prediction. Specifically, the model is trained to take the first generated image and input prompt as inputs and produce an artifact map. The training data include the RichHF-18K dataset and an additional 100K images manually annotated with bounding boxes (later transformed into map-style representations) to mark incorrectly generated regions according to the caption content.

**Correction Task**: This task equips the model with inpainting capabilities, enabling it to repair images based on the input caption and a masked image. Training leverages open-source datasets, including CC12M and SAM-1B, yielding 100K samples.

Following BrushNet's methodology (Ju et al., 2024), we generate both random and segmentation-based masks for training. Examples of these masks are shown in Figure 9.

## D DATASET CONSTRUCTION

### D.1 PRE-TRAINING DATA

In **Stage I** of the pre-training process, we utilized approximately 300 million image-text pairs to train the text-to-image (T2I) task. In **Stage II**, due to the increased resource demands associated with higher image resolutions, we improved the quality of images and captions through filtering and re-captioning techniques. Specifically, we employed an aesthetic model to curate high-quality images, and utilized the Qwen-VL-2.5 (Wang et al., 2024a) models to generate refined, high-quality captions

for these images. Ultimately, we collected and constructed a dataset comprising approximately 120 million samples for image generation task and 20 million samples for image understanding task.

## D.2 MCoT DATA

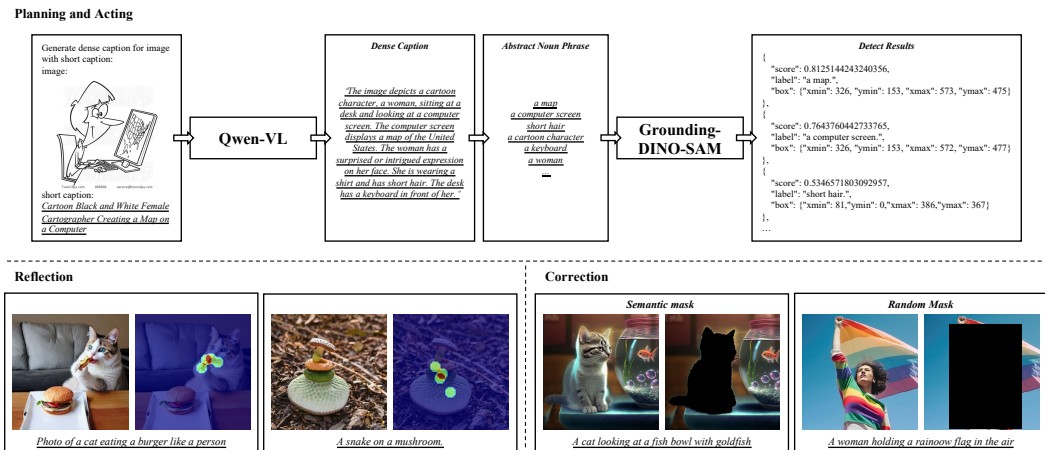

Figure 9: Data construction pipeline of MCoT training for image generation.

Figure 9 illustrates the complete data construction process of MCoT training for image generation, including data samples and the prompts used.

**For planning and acting tasks.** Image-caption pairs are expanded into image, caption, dense caption, bbox quadruples, where dense captions are augmented using Qwen-VL. Object unit phrases are parsed from dense captions and processed by Grounding-DINO-SAM (Liu et al., 2024d) to obtain corresponding bounding boxes for each object. The open-source dataset involved includes CC12M.

**For reflection tasks.** The RichHF-18K dataset, along with an additional 100K images annotated with bounding boxes for incorrect regions, are used. These images are manually annotated to identify bounding boxes of incorrectly generated objects according to corresponding caption contents. The model is trained to assess region defects and misalignment in the input generated image, represented by artifact map predictions, where brighter areas indicate greater need for correction.

**For correction tasks.** We adopt BrushNet's approach to generate random masks and segmentation masks for training. This trains the model to generate a repaired image based on the input caption and masked image.

## E FURTHER DISCUSSION ON FUNCTIONALITY CONFLICTS

Combining the two optimization objectives (next-token prediction loss and diffusion loss) within a single visual expert may introduce functional conflicts, as they pull visual representations toward different levels of the feature hierarchy. Specifically, next-token prediction loss for image understanding drives high-level text-semantic abstraction, while diffusion loss for image generation enforces low-level pixel fidelity. Moreover, such conflicts are difficult to observe directly, as they are implicit within the model's internal representations. We verify this through experiments (Table 7) and provide a more detailed discussion of the results. In the table, Dense architecture indicates that all tasks (text generation, image understanding, and image generation) share a single branch. Modality-Oriented architecture uses two branches, with one dedicated to text generation and the other to visual tasks (image understanding and generation). FoX employs three separate branches, with one for text generation and two for visual tasks, performing function splitting into image-understanding and image-generation branches. All architectures were trained under identical settings and validated on MS COCO, using CIDEr and FID to evaluate the model's image understanding and generation capabilities, respectively. The results show that FoX achieves significant improvements over the Modality-Oriented architecture in both image understanding and genera-

tion. This supports our claim that relying on a single expert to satisfy both objectives may lead to functional conflicts.

## F INFERENCE COST ANALYSIS AND TRADE-OFF STRATEGIES FOR MCoT

**Inference latency.** In both MCoT and T2I generation processes, the dominant latency comes from the diffusion forward pass, which scales linearly with the diffusion steps. The exact steps used in our experiments are:

- T2I baseline: 50 denoising steps
- MCoT: 50 (acting) + 30 (reflection) + 50 (correction) = 130 steps

Since the artifact map in the reflection step is relatively simple to generate, we use only 30 steps to reduce the latency. While the planning stage should also be included in principle, it generates only about 80 text tokens with KV-cache and contributes less than 5% of the total latency, so diffusion remains the dominant. Based on this, the theoretical latency increase is approximately $130/50 \approx 2.6\times$, and the final observed slowdown is around $2.8$–$3.0\times$ on an NVIDIA H20 GPU.

**Memory usage.** The GPU memory consumption of MCoT is around $1.2\times$ that of T2I, which is almost the same. The MCoT process can be viewed as three T2I procedures executed sequentially (planning-acting, reflection, and correction), so the GPU memory usage during each step is similar to that of T2I generation. MCoT only introduces a small amount of additional intermediate storage on the GPU, such as layouts, detailed captions, and artifact maps, which results in only a slight increase in memory usage.

While the inference latency and memory usage remain reasonable, our model significantly outperforms unified generative models such as JanusFlow and Show-o on GenEval, and even surpasses larger models such as SD3 with a 19% improvement in Attribute Binding, as reported in Table 1. Qualitatively, visual comparisons with current models in Figure 4 demonstrate that our generated images achieve precise alignment with complex compositional prompts while maintaining high visual quality. Additionally, the additional latency introduced by MCoT is typical, as similar overhead can also be observed in prior image-generation works (Jiang et al., 2025; Guo et al., 2025b). Improving the execution efficiency of the MCoT process remains a challenging problem, as reducing diffusion iterations without compromising compositional fidelity is challenging, making this an important direction for future exploration.

**Trade-off strategies.** To seek a performance-efficiency trade-off, we analyze in Table 15 how modifying MCoT pipeline stages affects latency and GenEval performance. We take the full MCoT pipeline as the baseline and report all latency values relative to its inference time. As shown in the first row, removing the planning stage and directly performing the following steps yields only a minor reduction in latency but leads to a substantial drop in performance. Similarly, removing reflection-correction sharply compromises the performance though it mitigates latency to some extent. Conversely, adding an additional reflection–correction cycle (third row) increases latency while yielding only marginal gains. Taken together, these results indicate that our chosen MCoT configuration achieves a well-balanced trade-off between efficiency and performance.

Table 15: **Trade-off Analysis of Performance and Latency for MCoT on GenEval.**

| Setting | Latency | Overall↑ | Single Obj. | Two Obj. | Counting | Colors | Position | Attr. Binding |
|---|---|---|---|---|---|---|---|---|
| MCoT without planning | 0.95× | 0.72 | 0.97 | 0.83 | 0.63 | 0.80 | 0.51 | 0.55 |
| MCoT without reflection–correction | 0.4× | 0.73 | 0.98 | 0.84 | 0.66 | 0.81 | 0.55 | 0.58 |
| MCoT | 1× | 0.77 | 0.99 | 0.86 | 0.71 | 0.82 | 0.60 | 0.64 |
| MCoT with two reflection–correction cycles | 1.7× | 0.78 | 0.99 | 0.88 | 0.73 | 0.82 | 0.61 | 0.65 |

## G BAD CASES OF THE REFLECTION AND MITIGATION STRATEGIES.

Failure cases could potentially occur in the reflection stage, but are mitigable. On one hand, when the reflection step highlights correct regions in the wrong image, it might not noticeably degrade the

final result, as the correct regions in the wrong image are usually preserved during correction. On the other hand, the issue could also arise for images that do not require any correction, essentially due to over-reflection. This can be mitigated by adjusting the training strategy of the reflection stage. Specifically, by including a proportion of images that require no correction, where the corresponding artifact maps are completely black, indicating no areas to highlight, the model learns to avoid unnecessary reflection, effectively alleviating over-reflection.

## H  LIMITATIONS

In this work, we focus on empowering a unified generative model by incorporating a multimodal chain of thought and then enhancing the image generation capabilities. We have not addressed more challenging fine-grained and customized image editing tasks, which would better demonstrate unified generative models' multimodal understanding, reasoning, and generation abilities. Exploring such tasks is part of our future research directions.

## I  BROADER IMPACT

This paper presents work whose goal is to advance the field of Machine Learning and Deep Learning. There are many potential societal consequences of our work, none of which we feel must be specifically highlighted here.

