# OpenReview forum: "Towards Enhanced Image Generation via Multi-Modal Chain of Thought in Unified Generative Models"
_ICLR.cc/2026/Conference — Submitted to ICLR 2026_

### Official Review · Reviewer_4655 · 2025-10-25

**Soundness:** 3
**Presentation:** 3
**Contribution:** 3
**Rating:** 6
**Confidence:** 3

**Summary:**

This paper addresses a critical limitation in contemporary unified generative models: their struggle with ​complex compositional image generation​ (e.g., multi-object scenes, spatial relationships). The authors identify that direct text-to-image (T2I) generation is insufficient for these challenges and propose a novel reasoning-based paradigm inspired by Chain-of-Thought (CoT).

The authors introduce ​FoXperts, an expert-parallel architecture that assigns experts based on function:1. A unified ​Linguistic Expert​ for text. 2. A dedicated ​Semantic Vision Expert​ for visual understanding tasks. 3. A dedicated ​Generative Vision Expert​ for visual generation tasks. Also, they propose ​MCoT, a four-step (Planning, Acting, Reflection, Correction) reasoning framework that emulates a human artistic workflow. The proposed model, ​FoX​ (1.3B parameters), demonstrates highly competitive performance across diverse benchmarks.

**Strengths:**

1. The idea of applying a Chain-of-Thought reasoning process to complex image generation is inspiring. The functionality-oriented expert architecture presents a fresh alternative to mainstream modality-oriented designs, effectively addressing a fundamental conflict in multimodal modeling.

2. The experimental evaluation is thorough. The paper validates its approach across a wide range of well-established benchmarks for both image generation and understanding, demonstrating the high performance of their model.

3. By moving from one-shot generation to a reasoned, multi-step process, it enhances the reliability and controllability of models for complex tasks. The proposed training paradigm is particularly significant as it provides a practical solution to a major data availability challenge.

**Weaknesses:**

1. What was the original intention behind using the VAE encoder for the "Image for Understand" component? Why not using a model with richer semantic features, like CLIP?

2. The overall training and inference process is multi-staged. What is the rationale for integrating and training these stages within a single model? For the same workflow, what are the advantages compared to using two expert models (e.g., a VLM for understanding and a generative model for creation)? For instance, can the understanding and generation tasks mutually enhance each other?

3. Although the total parameter of the model  is 1.3B, the entire image generation workflow is relatively long and time-consuming. How is the trade-off between performance and efficiency balanced?

4. Can the "reflection and correction" process be performed multiple times? If so, how does it affect the model's performance? Does the long-context issue impact this process? Could the authors provide some illustrative examples?

5. If an error occurs during the reflection stage, it will inevitably lead to mistakes in the subsequent correction. This may even result in a corrected image that is poorer than the initially generated one? How to prevent this issue as much as possible?

**Questions:**

Please refer to the weaknesses.

---

> ### Author Response · Authors · 2025-11-23
>
> Thank you very much for your positive and constructive feedback. We address each of your comments point-by-point below.
>
> ---
>
> **W1: Intention behind using the VAE encoder**
>
> **A1:** Thank you for your thoughtful comment. Compared with CLIP, whose visual representations are driven toward high-level text-semantic abstraction and thus lose some fine-grained details, the VAE encoder preserves fine-grained and structurally rich visual features. Such representations are beneficial for detailed image perception and then are helpful for MCoT components like the reflection stage. This choice is also consistent with prior works such as TransFusion [1] and LMFusion [2].
>
> [1] *Transfusion: Predict the Next Token and Diffuse Images with One Multi-Modal Mode,* ICLR 2025
>
> [2] *LMFusion: Adapting Pretrained Language Models for Multimodal Generation,* NeurIPS 2025
>
> ---
>
> **W2: Motivation for integrating all capabilities in one model**
>
> **A2:** Thank you for raising this important question. Pipelines that rely on multiple expert models can face the following issues, which simultaneously highlight the advantages of unified models and motivate our research within the unified generative modeling paradigm.
>
> - **Increased system complexity.** Pipelines with multiple expert models need to coordinate independently trained components, introducing significant deployment and engineering overhead. They often require extra infrastructure, fragile cross-model interfaces, and careful module synchronization, making them harder to extend and to perform joint post-training compared with a unified model.
> - **Limited multimodal integration.** Since each model in the pipeline operates independently, modalities are not jointly trained or tightly integrated. For example, standalone inpainting models are usually weak in the multimodal understanding needed to interpret prompts or global image context, which might lead to incoherent or semantically inconsistent corrections, unlike unified models that leverage cross-modal information end to end.
> - **Lossy communication and weak state continuity.** Modular pipelines exchange only final outputs and lack intermediate reasoning traces, discarding internal cues such as implicit layout cues and consistent style. This results in information loss and weaker stage coordination, often limiting downstream performance compared with unified models that preserve shared latent states across the workflow.
>
> As for the enhancement between understanding and generation, we illustrate this using the reflection–correction process. In the reflection step, stronger image understanding can help produce a more accurate artifact map, and in the correction step, better contextual understanding usually improves the coherence of generated images with the global context, showing how understanding can aid generation. As for the reverse direction, it is challenging to provide direct evidence, since such reinforcement would be likely implicit within the model’s internal dynamics and difficult to attribute explicitly. This remains a valuable direction for future investigation.

---

> ### Author Response · Authors · 2025-11-23
>
> **W3: The Inference pipeline is relatively long and time-consuming, and what is the trade-off**
>
> **A3:** We appreciate your insightful comment. Below we first provide the inference latency. In both MCoT and T2I, the dominant latency comes from the diffusion forward pass, which scales linearly with the diffusion steps. The exact steps used in our experiments are:
>
> - **T2I baseline:** 50 denoising steps
> - **MCoT:** 50 (acting) + 30 (reflection) + 50 (correction) = 130 steps
>
> Since the artifact map in the reflection step is relatively simple to generate, we use only 30 steps to reduce the latency. While the planning stage should also be included in principle, it generates only about 80 text tokens with KV-cache and contributes less than 5% of the total latency, so diffusion remains the dominant. Based on this, the theoretical latency increase is approximately $130/50 \approx 2.6\times$, and the final observed slowdown is around **2.8–3.0**× on an NVIDIA H20 GPU.
>
> While the inference latency remains reasonable, our model significantly outperforms Show-o and JanusFlow overall on GenEval, and even surpasses larger models such as SD3 with a 19% improvement in Attribute Binding, as reported in Tab. 1 of our paper. We also provide a visual comparison with current models in Fig. 4 of Appendix B, which shows that our generated images achieve precise alignment with complex compositional prompts while maintaining high visual quality. Additionally, the additional latency introduced by MCoT is typical, as similar overhead can also be observed in prior image-generation works [3,4]. Improving the execution efficiency of the MCoT process remains a challenging problem, as reducing diffusion iterations without compromising compositional fidelity is challenging, making this an important direction for future exploration.
>
> As for the trade-off between performance and efficiency, we further compare in the table below how adding or removing stages of the MCoT pipeline affects inference latency and GenEval performance. We take the full MCoT pipeline as the baseline and report all latency values relative to its inference time. As shown in the first row, removing the planning stage and directly performing the following steps yields only a minor reduction in latency but leads to a substantial drop in performance. Similarly, removing reflection-correction sharply compromises the performance though it mitigates latency to some extent. Conversely, adding an additional reflection–correction cycle (third row) increases latency while yielding only marginal gains. Taken together, these results indicate that our chosen MCoT configuration achieves a well-balanced trade-off between efficiency and performance.
>
> | Setting | Latency | Overall↑ | Single Obj. | Two Obj. | Counting | Colors | Position | Attr. Binding |
> | --- | --- | --- | --- | --- | --- | --- | --- | --- |
> | MCoT without planning | 0.95× | 0.72 | 0.97 | 0.83 | 0.63 | 0.80 | 0.51 | 0.55 |
> | MCoT without reflection–correction | 0.4× | 0.73 | 0.98 | 0.84 | 0.66 | 0.81 | 0.55 | 0.58 |
> | MCoT | 1× | 0.77 | 0.99 | 0.86 | 0.71 | 0.82 | 0.60 | 0.64 |
> | MCoT with two reflection–correction cycles | 1.7× | 0.78 | 0.99 | 0.88 | 0.73 | 0.82 | 0.61 | 0.65 |
>
> [3] *T2I-R1: Reinforcing Image Generation with Collaborative Semantic-level and Token-level CoT,* NeurIPS 2025
>
> [4] *Can We Generate Images with CoT? Let’s Verify and Reinforce Image Generation Step by Step,* CVPR 2025

---

> ### Author Response · Authors · 2025-11-23
>
> **W4: Can the "reflection and correction" process be performed multiple times**
>
> **A4:** Thank you for your valuable comment. Our reflection and correction steps can be performed multiple times. However, we find that a single reflection–correction cycle already provides a good balance between performance and efficiency. Executing multiple cycles yields only marginal gains while incurring additional inference cost. In the table below, we present a comparison on GenEval between running one versus two reflection–correction cycles. The results show that the improvements from a second cycle are relatively limited, with only minor changes across metrics. We also include qualitative examples for the two-cycle setting in [[LINK](https://anonymous.4open.science/r/2026-ICLR-rebuttal-reviewer-4655-BE72/figure-1.pdf)], where the first correction already meets most requirements and the second correction introduces only subtle adjustments.
>
> Regarding the long-context impact, our total token length is primarily determined by image resolution. At 256×256 resolution, the sequence length is only around 1k tokens, and at 1024×1024 resolution, it reaches about 10k tokens. With an average of roughly 5k tokens, this is well below the context limit of 32k tokens in Qwen2, so our model can handle the full context.
>
> | Setting | Overall↑ | Single Obj. | Two Obj. | Counting | Colors | Position | Attr. Binding |
> | --- | --- | --- | --- | --- | --- | --- | --- |
> | MCoT with one reflection–correction cycle | 0.77 | 0.99 | 0.86 | 0.71 | 0.82 | 0.60 | 0.64 |
> | MCoT with two reflection–correction cycles | 0.78 | 0.99 | 0.88 | 0.73 | 0.82 | 0.61 | 0.65 |
>
> ---
>
> **W5: Bad cases in the reflection stage and how to mitigate them**
>
> **A5:** Thank you for raising this valuable concern. Such failure cases could potentially occur but are mitigable. On one hand, when the reflection step highlights correct regions in the wrong image, it might not noticeably degrade the final result, as the correct regions in the wrong image are usually preserved during correction. On the other hand, the issue could also arise for images that do not require any correction, essentially due to over-reflection. This can be mitigated by adjusting the training strategy of the reflection stage. Specifically, by including a proportion of images that require no correction, where the corresponding artifact maps are completely black, indicating no areas to highlight, the model learns to avoid unnecessary reflection, effectively alleviating over-reflection.

---

> ### Author Response · Authors · 2025-11-27
>
> **Dear Reviewer 4655,**
>
> We hope you will excuse this follow-up message. With the discussion phase nearing its end, we wanted to gently confirm whether there are any outstanding matters we might assist in clarifying. We are deeply grateful for the attention you have already dedicated to our paper and remain at your disposal for any further guidance.
>
> Best regards,
>
> The Authors

---

> > ### Comment · Reviewer_4655 · 2025-11-28
> > **Reply to the authors**
> >
> > I thank the authors for their responses, which has addressed all of my concerns. I have no further questions at this time and will maintain my positive rating.

---

> > > ### Author Response · Authors · 2025-11-28
> > >
> > > **Dear Reviewer 4655,**
> > >
> > > We deeply appreciate that our responses have been able to **address all your concerns**. We are even more grateful for your full recognition of our idea, especially your remark that our idea of introducing CoT to complex image generation is inspiring, and our architecture presents a fresh alternative.
> > >
> > > Once again, thank you for your time and thoughtful comments, which have undoubtedly helped us improve our paper. We also **greatly appreciate your supportive positive recommendation. Wishing you all the best!**
> > >
> > > Best regards,
> > >
> > > The Authors

---

### Official Review · Reviewer_cMFY · 2025-10-27

**Soundness:** 3
**Presentation:** 3
**Contribution:** 2
**Rating:** 4
**Confidence:** 5

**Summary:**

This paper tackles the task of complex image generation by firstly introducing a unified generative model named FoX, which consists of FoXperts that disentangle experts by functions of generation and understanding. It also proposes an MCoT method to address image generation as a multi-step process of planning, acting, reflection, and correction, with a multi-task joint training paradigm to train the model without consistent multi-step data. Experiments are conducted on image generation and understanding benchmarks.

**Strengths:**

1. The model shows satisfying performance with a small number of parameters.
2. The generated images seem to effectively solve the complex image generation task, judging from the qualitative results.
3. The overall writing is clear and easy to follow.

**Weaknesses:**

1. The proposed method shows limited novelty compared with previous work. Assigning experts by functionality, the core contribution of FoX, is already introduced by BAGEL[1]. Using CoT in image generation is also present in works like T2I-R1[2], GoT-R1[3], Uni-CoT[4] but are not discussed in this paper. The main difference between them and the proposed MCoT lies in the layout planning, but this is somewhat confined to the compositional image generation task (e.g., T2I-CompBench) and cannot be extended to more general scenarios.
2. Benchmarks used in multimodal understanding are not sufficient. The experiments only consider MME-P, MMBench, and VQAv2. Commonly used ones like MMMU, MM-Vet, TextVQA, and InfoVQA are missing.
3. The model scale is still small (1.3B). It is unclear whether the proposed method can be scaled to larger models.


[1] Emerging Properties in Unified Multimodal Pretraining

[2] T2I-R1: Reinforcing Image Generation with Collaborative Semantic-level and Token-level CoT

[3] GoT-R1: Unleashing Reasoning Capability of MLLM for Visual Generation with Reinforcement Learning

[4] Uni-CoT: Towards Unified Chain-of-Thought ReasoningAcross Text and Vision

**Questions:**

1. Baselines in Table 2 are too old. What about recent models?
2. Can MCoT be extended to more general reasoning-related image generation benchmarks like WISE[5], T2I-ReasonBench[6] and PhyBench[7]?
3. Line 879: why is Semantic Visual Expert initialized from Generative Visual Expert? Is there an ablation on initializing from scratch or from Linguistic Expert (like in [8])?
4. Some writing issues:
- Line 170-172: I suppose these papers released in 2024 are not concurrent.
- Line 874: Qwne -> Qwen
- Table 9: No textual explanation. Better explain that the results are on T2I-CompBench.

[5] WISE: A World Knowledge-Informed Semantic Evaluation for Text-to-Image Generation

[6] T2I-ReasonBench: Benchmarking Reasoning-Informed Text-to-Image Generation

[7] PhyBench: A Physical Commonsense Benchmark for Evaluating Text-to-Image Models

[8] Mono-internvl: Pushing the boundaries of monolithic multimodal large language models with endogenous visual pre-training

---

> ### Author Response · Authors · 2025-11-23
>
> We sincerely appreciate your thoughtful comments. Below we address each point in detail.
>
> ---
>
> **W1: Novelty of FoXperts and MCoT**
>
> **A1:** Thank you for raising this valuable concern. We would like to restate our contribution more concisely and deeply.
>
> First, we illustrate the **innovation of FoXperts**.
>
> - FoXperts maintains **complete text–image modality separation while additionally introducing function-specific splitting for image understanding and generation.** On one hand, FoXperts with complete modality separation mitigates modality conflicts from a shared branch, as noted in MoT[1], enhancing both textual and visual capabilities. **On the other hand**, FoXperts mitigates the divergence between two optimization objectives: next-token prediction loss for image understanding, which leads to **high-level** text-semantic abstraction, and diffusion loss for image generation, which enforces **low-level** pixel fidelity. And combining them within a single visual expert may lead to potential functional conflicts, since these objectives pull visual representations toward different levels of the feature hierarchy.
> - In contrast, although BAGEL introduces two separate “understanding” and “generation” branches for functional assignment, their design remains with **incomplete modality separation**. Specifically,  its “understanding” branch processes text and image within the same expert, which reintroduces potential modality conflicts. As a result, placing any two modalities within the same expert can likely lead to performance degradation, which is empirically evidenced in previous work MoT[1] (please refer to Fig. 15 (a–e) of Sec. 4 in MoT), further suggesting that BAGEL can still suffer from potential modality-level conflicts that MoT has already identified.
>
> Secondly, we highlight **MCoT’s innovations**.
>
> - We begin by clarifying our innovations compared with the three papers you mentioned. Both T2I-R1 and GoT-R1 lack our **reflection-correction mechanism**, which is crucial as the model may still fail to render all parts perfectly in a single generation. This mechanism helps repair unresolved defects, such as Structural Incompleteness and Object Entanglement (Fig. 3), enhancing correctness and fidelity in complex compositional image generation. Quantitatively, T2I-R1 and GoT-R1 **perform worse** than our method on T2I-CompBench, as shown in the table below, despite their larger 7B architectures. Uni-CoT, published online in September, coinciding with our submission; thus, it was not discussed in our initial version. While Uni-CoT uses self-reflection via high-level text prompts mainly for style and semantic adjustment, we employ an artifact map that precisely identifies problematic regions and fine-grained defects, such as object entanglement, which are critical for complex compositional generation.
>
>
>     | Model | Params | Color↑ | Shape↑ | Texture↑ | Spatial↑ | Non-Spatial↑ | Complex↑ |
>     | --- | --- | --- | --- | --- | --- | --- | --- |
>     | T2I-R1 | 7B | 81.30 | 58.52 | 72.43 | 33.78 | 30.90 | 39.93 |
>     | GoT-R1 | 7B | 81.39 | 55.49 | 73.39 | 33.06 | 31.69 | 39.44 |
>     | FoX (Ours) | 1.3B | 82.37 | 59.81 | 74.21 | 35.71 | 34.19 | 42.78 |
> - We then discuss the applicability of layout planning. Layout planning is designed as a general mechanism that supports image generation tasks requiring layout design. It is therefore applicable not only to compositional image generation where prompts involve mixed conditions such as counting and position, **but also to many practical design tasks** **including advertisement and poster creation**, significantly reducing the human effort needed to create balanced and aesthetically pleasing layouts.
>
> Furthermore, our innovation is reflected in the **multi-task joint MCoT training paradigm**. Although reflection–correction steps improve image correctness and fidelity, **coordinating them** with planning–acting steps is **non-trivial**. Direct end-to-end MCoT training is hindered by the difficulty of collecting consistent multi-step data tuples, as creating a first “wrong” image aligned with context and final image while containing realistic errors is highly challenging. A naive solution is to generate such images using artificial errors, like randomly distorting regions, but this is too limited to capture the diversity of real image issues and may cause first-step outputs to always contain errors, undermining the goal of improving image quality through planning. To address this, we propose a multi-task joint training approach that splits the long sequence into simpler sub-tasks, without requiring coherent data across tasks, effectively avoiding the challenges of multi-step data collection.
>
> [1] *Mixture-of-Transformers: A Sparse and Scalable Architecture for Multi-Modal Foundation Models,* TMLR 2024

---

> ### Author Response · Authors · 2025-11-23
>
> **W2: Supplementary text understanding evaluation**
>
> **A2:** Following your insightful suggestion, we further evaluate FoX on three additional widely adopted and representative text understanding benchmarks, with results shown in the table below. The results indicate that FoX’s text understanding capabilities are comparable to current similar-scale models.
>
> | Model | Params | MMMU | MM-Vet | TextVQA |
> | --- | --- | --- | --- | --- |
> | TokenFlow-XL | 13B | 38.7 | 40.7 | - |
> | SEED-X | 13B | 35.6 | 43.0 | - |
> | Emu3-Chat | 8B | 31.6 | 37.2 | 64.7 |
> | Qwen2-VL | 7B | 54.1 | 62.0 | 84.3 |
> | ILLUME | 7B | 38.2 | 37.0 | - |
> | LLaVA-v1.5 | 7B | 35.4 | 31.1 | - |
> | VILA-U | 7B | - | 33.5 | 60.8 |
> | Chameleon | 7B | 22.4 | 8.3 | - |
> | BLIP-3 | 4B | 41.1 | - | - |
> | InternVL2 | 1.8B | 34.3 | 44.6 | - |
> | Gemini-Nano-1 | 1.8B | 26.3 | - | 62.5 |
> | LLaVA-v1.5-Phi-1.5 | 1.3B | 30.7 | - | - |
> | Show-o | 1.3B | 25.1 | - | - |
> | Janus | 1.3B | 30.5 | 34.3 | - |
> | JanusFlow | 1.3B | 29.3 | 30.9 | 55.5 |
> | FoX (Ours) | 1.3B | 31.3 | 33.7 | 57.2 |
>
> ---
>
> **W3: Scale up**
>
> **A3:** Thank you for raising this concern. We are running scale-up experiments with the larger Qwen2-1.5B backbone to validate our method’s generalizability. With the model size tripled, computational overhead has sharply increased, and results are still pending despite our best efforts to schedule additional GPU resources. We expect to obtain them in the next few days and will update the rebuttal immediately. In the meantime, we first provide a methodological explanation to show that our approach supports backbone-scale generalization.
>
> - For FoXperts, functionality disentanglement is implemented above the backbone level. Scaling the backbone up affects only feature extraction within each expert, leaving the functionality decomposition intact. This modularity is similar to that of MoE[2] systems, which naturally support larger backbones without modifying the routing mechanism. As a result, FoXperts is compatible with backbone scaling.
> - The MCoT framework is designed to be independent of a specific backbone. MCoT only requires that the model supports both text and image understanding and generation, which is already satisfied by unified generative models. Changing the backbone of the model does not compromise these capabilities. Therefore, the MCoT framework is compatible with backbone scaling.
>
> [2] *Outrageously Large Neural Networks: The Sparsely-Gated Mixture-of-Experts Layer*, 2017 ICLR
>
> ---
>
> **Q1:  New baselines for Table 2**
>
> **A4:** Following your important suggestion, we provide additional comparisons on GenEval with more recent models, including EMU3, Janus-Pro, and Show-O, as shown below. The results indicate that FoX continues to outperform current unified generative models. We will update Tab. 2 with this new comparison and also include the T2I-CompBench results for T2I-R1 and GoT-R1 as listed above in **A1 for W1**.
>
> | Model | Params | Color↑ | Shape↑ | Texture↑ | Spatial↑ | Non-Spatial↑ | Complex↑ |
> | --- | --- | --- | --- | --- | --- | --- | --- |
> | EMU3 | 7B | 75.44 | 57.06 | 71.64 | - | - | - |
> | Janus-Pro | 7B | 63.59 | 35.28 | 49.36 | 20.61 | 30.85 | 35.59 |
> | Show-o | 1.3B | 56 | 41 | 46 | 20 | 30 | 29 |
> | FoX (Ours) | 1.3B | 82.37 | 59.81 | 74.21 | 35.71 | 34.19 | 42.78 |
>
> ---
>
> **Q2: Other reasoning-related image generation benchmarks**
>
> **A5:** Thank you for your insightful comment. We further evaluated our model on the WISE benchmark. The results in the table below show that our model achieves performance comparable to other models of similar or even larger scale.
>
> | Model | Params | Cultural | Time | Space | Biology | Physics | Chemistry | Overall |
> | --- | --- | --- | --- | --- | --- | --- | --- | --- |
> | SD-3 | 12.7B | 0.42 | 0.44 | 0.48 | 0.39 | 0.47 | 0.29 | 0.42 |
> | T2I-R1 | 7B | 0.56 | 0.55 | 0.63 | 0.54 | 0.55 | 0.30 | 0.54 |
> | Emu3 | 7B | 0.34 | 0.45 | 0.48 | 0.41 | 0.45 | 0.27 | 0.39 |
> | Janus-Pro | 7B | 0.30 | 0.37 | 0.49 | 0.36 | 0.42 | 0.26 | 0.35 |
> | VILA-U | 7B | 0.26 | 0.33 | 0.37 | 0.35 | 0.39 | 0.23 | 0.31 |
> | SD-XL-base-0.9 | 3.4B | 0.43 | 0.48 | 0.47 | 0.44 | 0.45 | 0.27 | 0.43 |
> | Show-o | 1.3B | 0.28 | 0.36 | 0.40 | 0.23 | 0.33 | 0.22 | 0.30 |
> | Janus | 1.3B | 0.16 | 0.26 | 0.35 | 0.28 | 0.30 | 0.14 | 0.23 |
> | JanusFlow | 1.3B | 0.13 | 0.26 | 0.28 | 0.20 | 0.19 | 0.11 | 0.18 |
> | Janus-Pro | 1B | 0.20 | 0.28 | 0.45 | 0.24 | 0.32 | 0.16 | 0.26 |
> | FoX (Ours) | 1.3 B | 0.46 | 0.52 | 0.65 | 0.41 | 0.48 | 0.27 | 0.47 |

---

> > ### Author Response · Authors · 2025-11-25
> >
> > **Supplementary scaling-up results for W3**
> >
> > The table below presents our scale-up results using the larger Qwen2-1.5B backbone. Scaling the backbone from Qwen2-0.5B to Qwen2-1.5B yields consistent improvements across all metrics, which supports the scalability of our method. In particular, the Position score increases from 0.60 to 0.68 and the Attribute Binding score from 0.64 to 0.70, showing clear gains on challenging aspects.
> >
> > | Model | Overall↑ | Single Obj. | Two Obj. | Counting | Colors | Position | Attr. Binding |
> > | --- | --- | --- | --- | --- | --- | --- | --- |
> > | FoX-1.3B (Original) | 0.77 | 0.99 | 0.86 | 0.71 | 0.82 | 0.60 | 0.64 |
> > | FoX-4.2B (Scale-up) | 0.82 | 0.99 | 0.89 | 0.76 | 0.87 | 0.68 | 0.70 |

---

> ### Author Response · Authors · 2025-11-23
>
> **Q3: Semantic Visual Expert initialization**
>
> **A7:** Thank you for raising this insightful issue. Concretely, during pre-training, we first train the image-generation branch. When moving to the second-stage visual-understanding training, we initialize the Semantic Visual Expert using the weights of the Generative Visual Expert, as both belong to the image domain and are then more closely aligned than the text branch, which might better facilitate image understanding. We did not choose to train the Semantic Visual Expert from scratch due to higher training costs and potentially unstable optimization, as similarly noted in [3]. And we compared two different initialization strategies for the Semantic Visual Expert under identical training settings. Evaluation results on MS-COCO, using CIDEr for image understanding and FID for image generation. As shown in the table below, our model initialized with the Generative Visual Expert demonstrates better performance.
>
> | Setting | CIDEr↑ | FID↓ |
> | --- | --- | --- |
> | From Generative Visual Expert | 124.6 | 7.83 |
> | From Linguistic Expert | 122.1 | 7.97 |
>
> [3] *Mono-InternVL: Pushing the Boundaries of Monolithic Multimodal Large Language Models with Endogenous Visual Pre-training*, 2025 CVPR
>
> ---
>
> **Q4: Writing issues**
>
> **A8:** Thank you for raising these important issues. Following your suggestion, we have carefully reviewed the paper from start to finish and corrected all grammatical errors and missing information, including the issues you mentioned. The revised version has already been updated in our submission.

---

> ### Author Response · Authors · 2025-11-27
>
> **Dear Reviewer cMFY,**
>
> We respectfully submit this brief follow-up message. In light of the approaching deadline for the discussion stage, we wished to confirm whether there are any lingering concerns or questions we could help resolve. We are committed to fully addressing your feedback and would deeply appreciate any final thoughts you might be willing to share. Thank you for your thoughtful evaluation.
>
> Kind regards,
>
> The Authors

---

### Official Review · Reviewer_pLvw · 2025-10-30

**Soundness:** 3
**Presentation:** 3
**Contribution:** 3
**Rating:** 4
**Confidence:** 4

**Summary:**

The paper introduces FoXperts, a unified generative model with a Functionality-oriented eXperts architecture that mitigates function-domain conflicts in modality-oriented designs, while seamlessly integrating both understanding and generation across textual and visual modalities.

**Strengths:**

* The method is simple and easy to understand.
* MCoT splits complex drawing into four quick passes—plan, execute, reflect, refine—each targeting one goal so error drops round-by-round.
* FoXperts assigns “seeing” and “painting” to two separate vision experts, plus a language expert, eliminating internal conflict and giving later iterations a solid base.

**Weaknesses:**

* The paper mainly combines an expert architecture with a unified understanding–generation model. Technically, it divides the visual expert into two parts—semantic understanding and generation—which is rather straightforward. Essentially, it does not differ significantly from common mixture-of-experts models, so the innovation seems to lie more in integration than in architectural novelty.
* The proposed MCoT adopts a four-step process, which appears to be a standard approach in Chain-of-Thought (CoT) methods, without specific adaptations for image generation tasks.
* The idea of a planning → execution → reflection → revision process has already been reflected in existing CoT-based visual-language models (e.g., CoT-VLA), yet the paper lacks comparisons or discussions regarding these related works.
* Although the paper proposes a four-step process, it does not explain why such a complex procedure is necessary, as opposed to simpler two-stage (e.g., planning + generation) or three-stage designs.
* The paper claims that “a single visual expert leads to functional conflicts,” but provides neither experimental evidence nor theoretical analysis to demonstrate the existence or impact of such conflicts on performance.
* The paper argues that “direct T2I generation cannot handle complex compositional instructions,” yet current models like SD3 and DALL·E 3 already show strong performance in complex scene generation. The authors do not sufficiently justify why introducing CoT is essential rather than further optimizing existing generative models.

**Questions:**

See the weaknesses.

---

> ### Author Response · Authors · 2025-11-23
>
> We sincerely appreciate your thoughtful comments. Below we address each point in detail.
>
> ---
>
> **W1: Novelty of FoXperts**
>
> A1: Thank you for your insightful comment. We would like to first clarify our innovations compared with mixture-of-experts models (MoE). Our FoXperts **differs** from MoE fundamentally in both **architecture and routing**:
>
> - **Architecture:** FoXperts has **multiple branches**, each being **a separate, complete Transformer model**. In contrast, **MoE only has a single Transformer branch** and **modifies its internal structure**, such as replacing the feed-forward layer in each Transformer block with a set of parallel submodules.
> - **Routing:** FoXperts routes tokens **based on their modality**, so tokens from the same modality are consistently processed by the same Transformer branch. In contrast, **MoE does not consider modality** and routes each token individually to one or a few feed-forward submodules within the layer.
>
> Furthermore, FoXperts maintains **complete text–image modality separation while additionally introducing function-specific splitting for image understanding and generation,** mitigating both modality and functionality conflicts, and then enhancing both textual and visual capabilities.
>
> ---
>
> **W2&3: Novelty of MCoT**
>
> **A2:** Thank you for raising these important questions. We would like to restate our contribution more concisely and deeply.
>
> - We begin by highlighting our **specific adaptations for image generation tasks.** First, the planning step performs both caption and layout planning, emulating an artist sketching object positions and outlining details prior to painting. Second, the reflection step produces an artifact map that identifies problematic regions and reveals fine-grained defects such as object entanglement; meanwhile, the correction step repairs these issues according to global context, which improves the accuracy and fidelity of complex compositional generation. Moreover, to address the difficulty of collecting consistent multi-step MCoT data tuples for image generation, we introduce a multi-task joint training strategy that decomposes the long training sequence into four standard and simple sub-tasks, each corresponding to a step in MCoT. This paradigm does not require data from different tasks to share coherent relationships and also avoids the challenges of constructing multi-step MCoT tuples.
> - Next, we clarify our innovations compared with CoT-VLA, which has already been discussed in the Related Work of our paper. Here, we provide further clarification. First, CoT-VLA is specifically designed for embodied intelligence tasks in robotics. Concretely, its CoT procedure is tailored to decomposing actions for robotic tasks, making it fundamentally incompatible with our complex compositional image generation setting; thus, it was not adopted as a baseline in our paper. Moreover, our method is **clearly distinct from** CoT-VLA, as CoT-VLA **does not include our reflection-correction mechanism**, which is essential for complex image generation as noted above. We will provide a clearer discussion of CoT-VLA in the Related Work section and sincerely apologize for any misunderstanding caused.

---

> ### Author Response · Authors · 2025-11-23
>
> **W4: On the necessity of the four-step design**
>
> **A3:** We sincerely appreciate your valuable concerns. As the necessity of the four-step design is explained in the Reflection paragraph of Sec. 3.5.2, we restate it more concisely and deeply here. Despite the planning and acting steps improving image generation, the model may still fail to render all parts of an image perfectly in a single attempt. Extending the two-step planning-acting pipeline with additional reflection-correction steps further repairs remaining defects (Fig.3), significantly enhancing the final image quality. Furthermore, the quantitative ablations (Tab. 5 and 6 in our paper, reproduced below) validate that incorporating the reflection-correction steps contributes substantially to overall performance. While the two-step planning-acting pipeline already yields good results, the four-step design further improves overall performance on both GenEval and T2I-CompBench, particularly on the challenging Attribute Binding metric (0.58→0.64, relative improvement of 10.34%) and the Spatial metric (28.36→35.71, relative improvement of 25.92%). Since our reflection-correction operates as a continuous, combined process that cannot be separated, here we only compare the two-step and four-step pipelines.
>
> - Table 5: Ablation results on GenEval Benchmark for validating the effectiveness of MCoT.
>
>
>     | Setting | Single Obj. | Two Obj. | Counting | Colors | Position | Attr. Binding | Overall↑ |
>     | --- | --- | --- | --- | --- | --- | --- | --- |
>     | T2I Gen. Twice | 0.97 | 0.80 | 0.58 | 0.78 | 0.40 | 0.47 | 0.67 |
>     | MCoT Planning & Acting Only | 0.98 | 0.84 | 0.66 | 0.81 | 0.55 | 0.58 | 0.73 |
>     | MCoT Full Process | 0.99 | 0.86 | 0.71 | 0.82 | 0.60 | 0.64 | 0.77 |
> - Table 6: Ablation results on T2I-CompBench for validating the effectiveness of MCoT.
>
>
>     | Setting | Color | Shape | Texture | Spatial | Non-Spatial | Complex | Overall↑ |
>     | --- | --- | --- | --- | --- | --- | --- | --- |
>     | T2I Gen. Twice | 65.15 | 51.36 | 64.05 | 13.42 | 26.89 | 32.72 | 42.26 |
>     | MCoT Planning & Acting Only | 76.77 | 56.70 | 71.08 | 28.36 | 31.28 | 39.55 | 50.62 |
>     | MCoT Full Process | 82.37 | 59.81 | 74.21 | 35.71 | 34.19 | 42.78 | 54.85 |

---

> ### Author Response · Authors · 2025-11-23
>
> **W5: Additional clarification supporting functional conflicts**
>
> **A4:** Thank you for raising this insightful point. We would like to first restate the conflicts concisely and deeply. Combining the two optimization objectives (next-token prediction loss and diffusion loss) within a single visual expert may introduce functional conflicts, as they pull visual representations toward different levels of the feature hierarchy. Specifically, next-token prediction loss for image understanding drives high-level text-semantic abstraction, while diffusion loss for image generation enforces low-level pixel fidelity. Moreover, such conflicts are difficult to observe directly, as they are implicit within the model's internal representations. We verify this through experiments (Tab. 7 in our paper, reproduced below) and provide a more detailed discussion of the results.
>
> - Table 7: Ablation of FoXperts
>
>
>     | Architecture | CIDEr↑ | FID↓ |
>     | --- | --- | --- |
>     | Dense | 116.2 | 11.3 |
>     | Modality-Oriented | 121.1 | 9.56 |
>     | FoX (Ours) | 126.5 | 7.24 |
>
> In the table, Dense architecture indicates that all tasks (text generation, image understanding, and image generation) share a single branch. Modality-Oriented architecture uses two branches, with one dedicated to text generation and the other to visual tasks (image understanding and generation). FoX employs three separate branches, with one for text generation and two for visual tasks, performing function splitting into image-understanding and image-generation branches. All architectures were trained under identical settings and validated on MS COCO, using CIDEr and FID to evaluate the model's image understanding and generation capabilities, respectively. The results show that FoX achieves significant improvements over the Modality-Oriented architecture in both image understanding and generation. This supports our claim that relying on a single expert to satisfy both objectives may lead to functional conflicts.
>
> ---
>
> **W6: Why is introducing CoT essential rather than further optimizing existing generative models**
>
> **A5:** We sincerely appreciate your thoughtful comment. We sincerely apologize for any confusion caused. We meant to convey that, in unified generative models like Show-o, simply using a text-to-image (T2I) process may not fully handle complex compositional instructions. This does not imply that all current unimodal T2I models, such as SD3, are incapable of handling such tasks.
>
> Second, in Tab. 1(partially reproduced below) in our paper, we have compared FoX with SD3 and DALL·E 3 on complex image generation tasks. Here, we further discuss the performance of SD3 and DALL·E 3. Our FoX demonstrates clear overall improvements, with +9% over SD3 and +10% over DALL·E 3 on GenEval, especially on the challenging Attribute Binding metric (+20% and +19%, respectively). And our paper also includes qualitative comparisons with SD3 (Fig. 4), showing issues such as Object Defects in two-object tasks, as well as Spatial Confusion in position tasks, indicating SD3 still struggles with complex image generation. To further evaluate the practical performance of SD3 and DALL·E 3, we test the open-source and widely used SD3 on challenging examples in Fig. 1 containing three or four objects—a scenario commonly encountered in practice—and compare the results with ours. As shown in [[LINK](https://anonymous.4open.science/r/2026-ICLR-rebuttal-reviewer-pLvw-84E7/figure-1.pdf)], the issues for SD3 are even more pronounced, confirming that complex compositional tasks remain challenging. Importantly, our images maintain strong prompt alignment and high fidelity, demonstrating that our MCoT framework effectively enhances the model’s ability to solve these tasks.
>
> - Table 1: Comparison of enhanced image generation quality on GenEval.
>
>
>     | Model | Param | Overall↑ | Single Obj. | Two Obj. | Counting | Colors | Position | Attr. Binding |
>     | --- | --- | --- | --- | --- | --- | --- | --- | --- |
>     | SD3 | 12.7B | 0.68 | 0.98 | 0.84 | 0.66 | 0.74 | 0.40 | 0.43 |
>     | DALL-E 3 | -- | 0.67 | 0.96 | 0.87 | 0.47 | 0.83 | 0.43 | 0.45 |
>     | FoX (Ours) | 1.3B | 0.77 | 0.99 | 0.86 | 0.71 | 0.82 | 0.60 | 0.64 |
>
> Third, we would like to clarify the necessity of introducing CoT. Optimizing generative models like SD3 and DALL·E 3 often comes at a tremendous cost, such as collecting and cleaning large volumes of high-quality data and training at a massive scale. In contrast, introducing a CoT framework generally requires less data and lower computational overhead for fine-tuning, yet can still enhance model capabilities. As discussed above, our model with CoT outperforms SD3 and DALL·E 3 on complex image generation tasks. Although further optimizing models like SD3 remains challenging and costly, it is still a valuable research direction.

---

> ### Author Response · Authors · 2025-11-27
>
> **Dear Reviewer pLvw,**
>
> We apologize for this additional correspondence. As the discussion window draws to a close, we respectfully wish to ensure that we have thoroughly addressed all aspects of your assessment. If there are any remaining points that require further elaboration, we would be most grateful for the opportunity to clarify them. We sincerely value your insights.
>
> Warm regards,
>
> The Authors

---

### Official Review · Reviewer_Zoat · 2025-10-31

**Soundness:** 3
**Presentation:** 3
**Contribution:** 3
**Rating:** 4
**Confidence:** 4

**Summary:**

The study introduces a unified generative model that improves complex image creation through a functionality-oriented expert design and a multimodal chain of thought process. The model separates visual understanding and generation to strengthen both abilities and follows a four-step reasoning workflow of planning, acting, reflection, and correction. A multi-task training scheme enables each step to be learned independently without costly supervision. Experiments on several benchmarks show clear gains in compositional accuracy and visual quality, highlighting the value of combining functional expert design with stepwise reasoning for better image generation.

**Strengths:**

1. The separation of generation and understanding experts brings performance gains, which shows potential in unified MLLMs.
2. The multi-task joint training paradigm enables efficient learning of each reasoning step without requiring expensive multi-step supervision, making the approach scalable in terms of data efficiency.
3. The proposed FoX achieves best results on most generation benchmarks, which shows the effectiveness for this framework.

**Weaknesses:**

1. All experiments are conducted using the Qwen2 0.5B backbone without comparisons across larger scales or alternative architectures, leaving uncertainty about the method’s scalability and general applicability.
2. The proposed MCoT framework introduces multiple reasoning steps (planning, acting, reflection, correction), which likely increase inference time and computational cost compared with direct text-to-image generation.

**Questions:**

1. The multi-step MCoT process (planning, acting, reflection, correction) likely increases inference time. Can the authors provide quantitative comparisons of inference latency and memory usage versus baseline text-to-image generation?
2. Could the authors clarify whether the proposed FoX and MCoT framework can generalize to other backbone architectures or larger models beyond Qwen2 0.5B?

---

> ### Author Response · Authors · 2025-11-23
>
> Thank you very much for the constructive feedback.  Below, we address each comment point-by-point.
>
> ---
>
> **W1&Q2: Backbone scaling and architecture generalization**
>
> **A1:** Thank you for raising this concern. We are running scale-up experiments with the larger Qwen2-1.5B backbone to validate our method’s generalizability. With the model size tripled, computational overhead has sharply increased, and results are still pending despite our best efforts to schedule additional GPU resources. We expect to obtain them in the next few days and will update the rebuttal immediately.
>
> In the meantime, we agree that adopting a stronger architecture, such as the more recent Qwen3, could potentially improve performance, and we would also like to conduct validation across different architectures. Nevertheless, scaling experiments are already burdensome, as training the current FoX (1.3B) requires 128 GPUs for 14 days. Therefore, we first provide a methodological explanation to show that our approach supports both scaling and swapping across different architectures.
>
> - For FoXperts, functionality disentanglement is implemented above the backbone level. Changing the backbone, either by scaling or replacing it, affects only feature extraction within each expert, leaving the functionality decomposition intact. This modularity is similar to that of MoE[1] systems, which naturally support larger or different backbones without modifying the routing mechanism. As a result, FoXperts is compatible with both backbone scaling and architecture generalization.
> - The MCoT framework is designed to be independent of a specific backbone. MCoT only requires that the model supports both text and image understanding and generation capabilities, which is already satisfied by unified generative models. Changing the backbone of the model does not compromise these capabilities. Therefore, the MCoT framework is compatible with backbone scaling and swapping.
>
> [1] *Outrageously Large Neural Networks: The Sparsely-Gated Mixture-of-Experts Layer*, 2017 ICLR
>
> ---
>
> **W2&Q1: Inference cost of MCoT**
>
> **A2:** We appreciate your insightful comment. Below we first provide the inference latency. In both MCoT and T2I, the dominant latency comes from the diffusion forward pass, which scales linearly with the diffusion steps. The exact steps used in our experiments are:
>
> - **T2I baseline:** 50 denoising steps
> - **MCoT:** 50 (acting) + 30 (reflection) + 50 (correction) = 130 steps
>
> Since the artifact map in the reflection step is relatively simple to generate, we use only 30 steps to reduce the latency. While the planning stage should also be included in principle, it generates only about 80 text tokens with KV-cache and contributes less than 5% of the total latency, so diffusion remains the dominant. Based on this, the theoretical latency increase is approximately $130/50 \approx 2.6\times$, and the final observed slowdown is around **2.8–3.0**× on an NVIDIA H20 GPU.
>
> As for memory usage, the GPU memory consumption of MCoT is around **1.2×** that of T2I, which is almost the same. The MCoT process can be viewed as three T2I procedures executed sequentially (planning-acting, reflection, and correction), so the GPU memory usage during each step is similar to that of T2I. MCoT only introduces a small amount of additional intermediate storage on the GPU, such as layouts, detailed captions, and artifact maps, which results in only a slight increase in memory usage.
>
> While the inference latency and memory usage remain reasonable, our model significantly outperforms Show-o and JanusFlow overall on GenEval, and even surpasses larger models such as SD3 with a 19% improvement in Attribute Binding, as reported in Tab. 1 of our paper. We also provide a visual comparison with current models in Fig. 4 of Appendix B, which shows that our generated images achieve precise alignment with complex compositional prompts while maintaining high visual quality. Additionally, the additional latency introduced by MCoT is typical, as similar overhead can also be observed in prior image-generation works [2,3]. Improving the execution efficiency of the MCoT process remains a challenging problem, as reducing diffusion iterations without compromising compositional fidelity is challenging, making this an important direction for future exploration.
>
> [2] *T2I-R1: Reinforcing Image Generation with Collaborative Semantic-level and Token-level CoT,* NeurIPS 2025
>
> [3] *Can We Generate Images with CoT? Let’s Verify and Reinforce Image Generation Step by Step,* CVPR 2025

---

> > ### Author Response · Authors · 2025-11-25
> >
> > **Supplementary scaling-up results for W1&Q2**
> >
> > The table below presents our scale-up results using the larger Qwen2-1.5B backbone. Scaling the backbone from Qwen2-0.5B to Qwen2-1.5B yields consistent improvements across all metrics, which supports the scalability of our method. In particular, the Position score increases from 0.60 to 0.68 and the Attribute Binding score from 0.64 to 0.70, showing clear gains on challenging aspects.
> >
> > | Model | Overall↑ | Single Obj. | Two Obj. | Counting | Colors | Position | Attr. Binding |
> > | --- | --- | --- | --- | --- | --- | --- | --- |
> > | FoX-1.3B (Original) | 0.77 | 0.99 | 0.86 | 0.71 | 0.82 | 0.60 | 0.64 |
> > | FoX-4.2B (Scale-up) | 0.82 | 0.99 | 0.89 | 0.76 | 0.87 | 0.68 | 0.70 |

---

> ### Author Response · Authors · 2025-11-27
>
> **Dear Reviewer Zoat,**
>
> We intend to respectfully follow up as the discussion phase draws to a close. We wish to ensure that we have adequately addressed your previous comments and to inquire if there are any outstanding queries we might further clarify. We remain at your disposal should you require any additional information. We sincerely appreciate your time and guidance.
>
> Best regards,
>
> The Authors

---

> > ### Comment · Reviewer_Zoat · 2025-11-28
> >
> > Thanks for the reply from the authors, which mostly resolves my concern. I will raise my score accordingly.

---

> > > ### Author Response · Authors · 2025-11-28
> > >
> > > **Dear Reviewer Zoat,**
> > >
> > > We **sincerely thank you** for your meticulous review of our clarification and for **offering an increase in your score!** We are truly encouraged and excited by your feedback!
> > >
> > > Once again, thank you for your time, thoughtful, and insightful comments, which have undoubtedly helped us improve our paper. We also greatly appreciate your significant support through the score increase. **Wishing you all the best!**
> > >
> > > Best regards,
> > >
> > > The Authors

---

### Meta-Review · Area_Chair_jBwu · 2026-01-06

**Summary:**

The major concerns raised by the reviewers include:
1. Insufficient evaluation, including experiments with different baseline models, ablation study with different model scale, insufficient comparison with baseline models.
2. The proposed method introduce higher computational cost yet lacks related analysis
3. Limited novelty in model design and CoT approach, and the motivation and benefits are not clearly presented. Also the literature review is incomplete, leaving the difference with related works and the contribution unclear.
4. The benefit of using a single model for CoT is not clearly justified

**Reviewer Concerns:**

1. The rebuttal provides additional experiments and partially addresses concerns regarding the evaluation. However, it remains insufficient to demonstrate the generalizability of the proposed method across different baseline models, which would help strengthen the paper. In addition, comparisons with more advanced text-to-image models are still needed to better justify the contribution of the proposed approach.
2. The rebuttal resolves the concerns regarding computational cost.
3. Concerns related to novelty and the literature review are partially addressed; however, the literature review in the paper should be further improved to clearly introduce the technical contributions.
4. The argument for using a single model is not substantial and does not resolve the corresponding concern.

**Reviewer Scores:**

1. While some concerns regarding the evaluation may be resolved through further discussion, other concerns require substantial additional work and fall outside the scope of the rebuttal and discussion wouldn't help.
2. Reviewers for whom computational cost is the primary concern are likely to increase their ratings.
3. Further discussion of the novelty is unlikely to lead to a higher rating.
4. The reviewer who raised concerns about the benefit of using a single model did not increase their rating, despite having no further questions.

---

### Decision · Program_Chairs · 2026-01-26

Reject